# Physics-based broadband characterization of weak earthquakes

František Gallovič [1] ✉, Sara Sgobba[2] & Ľubica Valentová K. [1]

Analyses of abundant small earthquakes have the potential to map the Earth's small-scale stress and frictional properties. However, standard seismological characterizations lack the resolution to capture the physical complexity of earthquake rupture, particularly its spatial and temporal heterogeneity. Advanced dynamic rupture models, which integrate elastodynamic simulations with frictional laws, have so far been applied to large earthquakes. Here, we develop a Bayesian dynamic inversion from station-specific (apparent) source spectra, extending up to 25 Hz, for slip-weakening friction, assuming heterogeneous parameterization along a finite-extent planar fault with resolution down to ~100-m scale. The approach is demonstrated on two Mw~4 earthquakes in Central Italy with distinct spectral behavior, one directive and one nondirective. Results show that the inversion resolves key mean source parameters, pinpoints possible pitfalls in standard estimates, and infers a power-law decay of stress and friction heterogeneity spectra down to the smallest scales. Such advanced studies promise to unravel so-far elusive small-scale characteristics of earthquake ruptures.

Earthquakes are presumably self-similar phenomena, as suggested by observations indicating that stress drop remains independent of earthquake magnitude across a broad range of event sizes[1,2]. While large earthquakes are routinely studied by detailed kinematic and dynamic slip inversions thanks to seismological and geodetic observations at short distances, analyses of smaller events typically rely on fitting spectral source models to processed P- or S-wave traces. Rare kinematic slip inversions of small earthquakes generally demonstrate spatial variability in slip distribution like that well known for large earthquakes[3–5]. A similar conclusion was reached by Yoshida[6] from the observed complexity of the shapes of the direct P waves of small events observed in near-source borehole data in Japan.

The temporal evolution of earthquake source radiation as observed by various receivers is described by an apparent (station-dependent) source time function (ASTF)[1]. ASTF or their amplitude spectra (so-called apparent spectra, AS) are commonly used to estimate earthquake source characteristics (rupture size, stress drop, etc.) under the assumption of simplified[7,8] source models. These models generally assume an omega-squared source spectrum with a corner

frequency depending on the seismic moment and stress drop. However, this is an oversimplification because the corner frequency (as well as the whole spectrum) generally depends on the shape of the ASTF[9–12]. Empirical source spectra often exhibit high-frequency decay that deviates from the classical omega-squared model. This behavior is sometimes characterized using the so-called κ-source parameter, which quantifies attenuation at high frequencies[13].

Due to the rupture directivity effect, the ASTF and AS can exhibit spatially strongly varying apparent duration and corner frequencies, respectively[14–18]. Azimuthal variations in corner frequency and spectral decay from smooth dynamic ruptures have been extensively explored by Kaneko and Shearer[19], who suggested that properly accounting for the spatial dependence of source spectral properties could provide valuable constraints on source parameters. Yoshida[20] demonstrated that asymmetrical rupture propagation may result in biased stress drop estimates if the directivity effect on the source spectra is neglected. Pacor et al.[16] and Colavitti et al.[14] demonstrated that the empirical AS as a whole (not only the corner frequency) can exhibit complex azimuthal dependence even for weak directive events in

[1]Department of Geophysics, Faculty of Mathematics and Physics, Charles University, Prague, Czech Republic. [2]Istituto Nazionale di Geofisica e Vulcanologia, Milan, Italy. ✉e-mail: frantisek.gallovic@matfyz.cuni.cz

Central Italy, assigning it tentatively to rupture process heterogeneity. Schliwa and Gabriel[21] demonstrated by numerical simulations that the local rupture heterogeneities are imprinted in the spatial distributions of apparent corner frequencies. Calderoni and Abercrombie[22] point out the trade-off between high stress drop and increased rupture complexity, as both factors contribute to stronger high-frequency ground motions. Nevertheless, the stress drop and rupture complexity likely affect the radiated wavefield with distinct azimuthal and frequency dependence. Therefore, an inversion utilizing a suitable finite-extent physics-based rupture model might resolve and quantify such competing source effects.

In dynamic source inversions, frictional parameters on a fault are inferred from observed waveforms, obtaining a rupture model constrained by both the data and the governing physics (elastodynamic equation and friction law). Bayesian dynamic rupture inversions of regional waveforms have already been performed for several large events[23–26]. The waveform inversions are generally limited by the accuracy of Green's functions describing the path and site effects, restricting their application to low-frequency waveforms. As a result, they are unsuitable for analyzing small earthquakes and unraveling small-scale source properties. Therefore, an inversion of AS corrected for path and site effects in a broad frequency range represents a tempting alternative to constrain a dynamic model over a wide range of scales. Yet, up to now, there has been no attempt to invert the AS for dynamic rupture parameters directly.

Although earthquake ruptures are generally affected by many physical source phenomena (heat production, off-fault fracturing and plastic deformation, fault non-planarity, inhomogeneous fault rheology, etc.), here we attempt to use a relatively simple dynamic model yet with generally heterogeneous prestress and slip-weakening parameters along a planar fault embedded in a homogeneous space. Whereas theoretical studies often employ smooth models with strong barriers to reproduce high-frequency spectra, we instead adopt a heterogeneous distribution of dynamic parameters to capture rupture complexity. The source heterogeneity is suggested across multiple scales in, e.g., laboratory experiments[27–29], field observations of natural faults[30–32] or strong motion simulations[33–38]. Note that, unlike kinematic source models, the rupture dynamics guarantee that the inferred models are physically consistent, thus serving as a physical constraint in the inverse modeling.

As input data for the inversion, we use apparent (station-dependent) source spectra (AS) up to 25 Hz obtained by the Generalized Inversion Technique (GIT[39–42]). GIT is a non-parametric approach that decomposes recorded spectra into source, path, and site contributions, thereby providing robust corrections for path and site effects and yielding reliable source spectra for rupture characterization. Since the spectra do not include the phase information, we account for the associated uncertainties within the Bayesian framework. To our knowledge, a dynamic inversion of apparent amplitude spectra has not been attempted previously; therefore, we perform a synthetic test to guide the interpretation of the real-data inversion.

In this work, we focus on small M4 earthquakes in Central Italy to exploit the full frequency range while making the dynamic simulation computationally tractable. The Central Italy region, one of the most seismically active in Italy, has produced tens of thousands of well-recorded events (by up to hundreds of stations) and thus has been subject to numerous source studies addressing stress drop scaling, though all assume simplified source spectral model[22,43–45]. The present study develops a methodology that relaxes the assumption of simplified source radiation while employing similar input data. We select two events yet with distinct AS behavior to be compared in terms of their source characteristics and variability within the Bayesian framework. Assuming a heterogeneous distribution of model parameters, we recover complex rupture propagation down to the smallest scales ~100 m that is both physics and data constrained. In addition, we

compare our results with those determined by common seismological practice based on standard source spectral models and finally, investigate the power spectral densities of the heterogeneities in the inferred dynamic rupture parameters.

## Results

The two example events occurred during 2016-2017 Central Italy sequence: an Mw4.2 aftershock of the 2016-10-30 Norcia mainshock and an Mw4.5 aftershock of the 2016-08-24 Amatrice earthquake. Figure 1b shows the AS for the two events as a function of frequency with color-coded station azimuth to emphasize the potential systematic dependence and spatial distribution of the AS amplitudes. It also highlights different AS variability between the two events over the whole frequency range. Figure 1a then shows the deviation of the AS from the average at an example frequency of 10 Hz in a map view, confirming no systematic dependence on the distance from the source. The azimuthal coverage is good with 16° and 35° gaps for the directive and nondirective events, respectively, yet with a prevalence of stations in the NW-SE direction (see the rose diagram in Fig. 1a), following the orientation of the Apennines. Visual inspection of Fig. 1 suggests that the Mw4.2 earthquake exhibits a strong directivity effect towards the south, while no spatial trend is visible for the Mw4.5 event.

### Best rupture models

We examine the maximum a-posteriori (MAP) models of the two events (i.e., having the lowest misfit). Figure 2a displays their kinematic characteristics (slip, stress drop, rupture time, and rupture speed), and Supplementary Movies S1, S2 show their rupture propagation. Models of both earthquakes are complex down to the smallest scales, featuring locally high stress drops and numerous local decelerations and accelerations of the rupture up to super-shear speeds. As expected, the directive Mw4.2 event has a more asymmetric rupture due to the nucleation point being closer to one side of the ruptured area, unlike the nondirective Mw4.5 event. The Mw4.5 earthquake ruptured a larger portion of the fault and had larger stress-drop peaks than the Mw4.2 event. Figure 2b displays dynamic parameters confirming the significant degree of spatial heterogeneity of the dynamic parameters, which translates to the heterogeneity of the slip distributions and rupture speeds described above.

Figure 3a displays the bias between the observed and synthetic AS, evaluated as the natural logarithm of their ratio. The average bias being close to zero in the entire frequency range demonstrates that the AS's overall level and azimuthal dependence are fitted well for both the directive and nondirective events. This is particularly interesting considering the distinct azimuthal and frequency dependence of the GIT AS of the two events (Fig. 1), see also the synthetic spectra in Supplementary Fig. S1. The standard deviation of the bias is approximately frequency-independent and similar for both events (0.45-0.46). The bias can be further explored in a map view plotted for selected frequencies in Fig. 3b, demonstrating very weak azimuthal and distance dependence. Note that the -30% underestimation at the lowest frequencies is related to the seismic moment $M_0$; models with higher $M_0$ overestimate the data by a similar amount in that frequency range.

### Ensemble rupture properties

Since the dynamic inversion of AS amplitude spectra has not been performed previously, we must properly inspect the uncertainties revealed by the Bayesian framework, here represented by the rupture model ensembles. In particular, we obtained 4745 and 3667 dynamic rupture models for the Mw4.2 and Mw4.5 events, respectively.

Figure 4a shows moment rate (MR) functions calculated as integrals of the slip rates along the fault multiplied by the shear modulus calculated from the medium properties (Table 1). The MRs are color-coded by the posterior PDF values. The MAP model is shown in black and has ~1 s duration for both events with a weak onset due to the

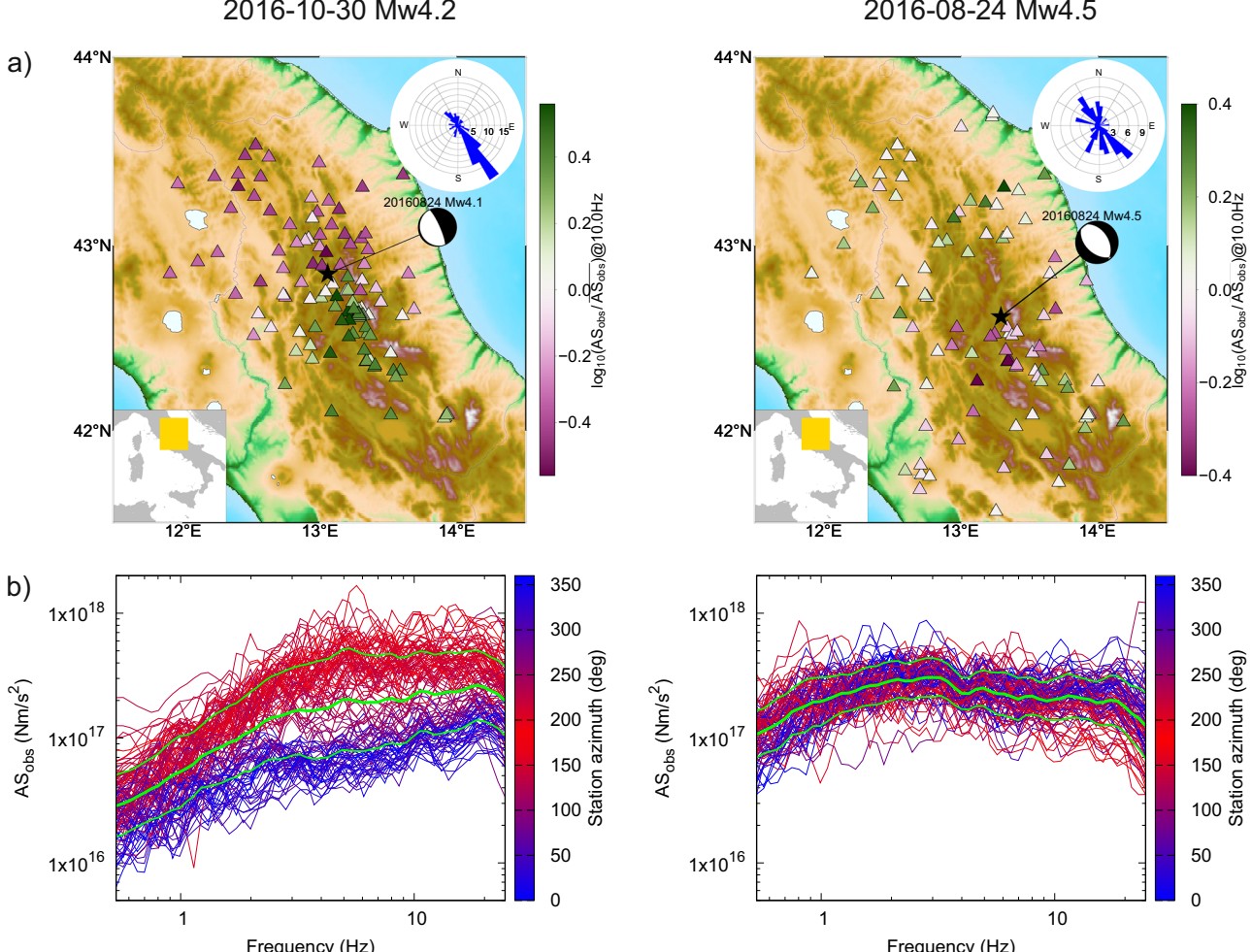

**Fig. 1 | Apparent spectra (AS) derived by the Generalized Inversion Technique (GIT) employed in the inversion of the two studied events (columns). a** Station distribution (triangles), epicenter location (star), and centroid moment tensor (beachball) of events 2016-10-30 Mw4.2 (left) with southward directivity and 2016-08-24 Mw4.5 (right) with no directivity. The stations are color-coded by the difference (in log) from the mean value over all stations at 10 Hz. The bottom-left insets show a map of Italy with the study area in yellow. The top-right rose diagrams show the azimuthal histograms of the station distributions. **b** GIT-derived AS from the stations shown in the respective map in panel a, color-coded by the station azimuth. Green spectra correspond to the mean and standard deviations. Note the remarkable difference between the two events in the azimuthal variability of their spectra, demonstrating the prominent directivity effect of the Mw4.2 event as seen in the map in panel (**a**). Note also that neither of the average spectra exhibits a perfect omega-squared decay.

prescribed weak nucleation (see "Methods"). The temporal spread of the MRs is relatively wide, including cases with delayed nucleation, mainly because the amplitude spectra lack explicit temporal information.

In addition, we compare the ensemble amplitude spectra of the MR second derivative (so-called moment jerk) with the spectrum averaged over empirical AS and its uncertainty (cf. green curve in Fig. 1b). In empirical studies, the station-averaged AS is considered to represent the MR spectrum to derive source characteristics. Figure 4 shows that while this assumption is reasonable for the nondirective circular Mw4.5 rupture, where the observed average spectrum matches the ensemble MR spectra within uncertainty, it may be disputable for the directive Mw4.2 event, where the average AS overestimates the MR spectrum. This affects the derived source characteristics of the earthquakes, as discussed later.

To reveal stable slip features, Fig. 4b displays the average slip distribution (in color scale) for the two events. Both slip models attain relatively simple, smooth, quasi-circular shapes. This suggests that the localized peaks in slip seen in the MAP models in Fig. 2a vary in position over the ensemble models, and the ensemble averaging smooths them out. Figure 4b also shows the variability of the rupture extent

over the ensemble in terms of mean and one-sigma standard deviation contours of slip. We note that the individual models exhibit diverse shapes, which produce circular distribution in the ensemble averaging.

Figure 5 shows the ensemble averages of the dynamic rupture properties along the fault and their standard deviations. Both the ensemble-averaged prestress and friction drop are smoothed out, being approximately constant within the average rupture area, with increased variability towards its border due to the variability of the individual rupture extents. The constant values are similar for both events, the prestress and friction drop being in the range of 3-4 MPa and 0.08-0.09 (i.e., the strength of 8-9 MPa), respectively. The slip-weakening distance of both events increases with distance from the nucleation, $\rho$, approximately as $D_c(\rho) = 0.35[cm] + 3.8\rho[km]$ and $D_c(\rho) = 0.5[cm] + 4.1\rho[km]$ for the Mw4.2 and Mw4.5 event, respectively. We would like to emphasize that the smooth average dynamic model (if able to rupture) would be unable to fit the observed AS data (see Materials and Methods and cf. Supplementary Fig. S2).

Figure 6 shows histograms of the misfit in the ensembles for the two events (Mw4.5 in red and Mw4.2 in blue). The misfit attains slightly

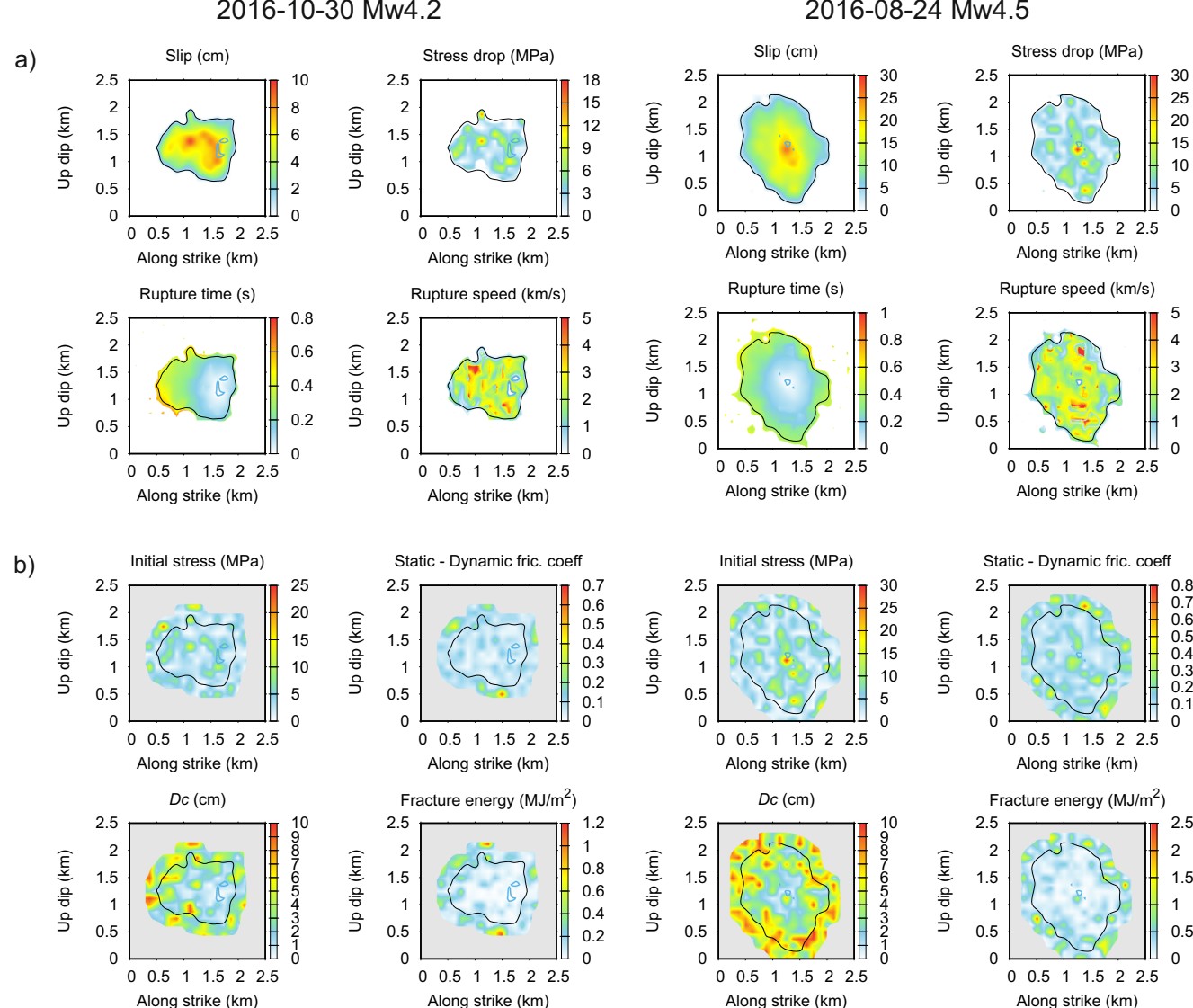

**Fig. 2 | Rupture properties of the maximum a-posteriori (MAP) models for the two studied events (columns). a** Kinematic and (**b**) dynamic rupture parameters. Note that the fracture energy is calculated from $D_c$ and friction drop values. Blue contours outline the nucleation area (i.e., the patch with negative strength excess), and black lines contour the slip distribution at 10% of the respective maximum. Part of the model that is unconstrained by the data is masked by gray.

lower values for the Mw4.5 rupture, which can be attributed to the lower number of stations (93) than for the Mw4.2 event (117). The similar misfit values confirm a similar fit of the data despite their very different azimuthal dependence and variability (Fig. 1b). Figure 6 also displays the distribution of seismic moments $M_0$ and the assumed prior distributions. The posterior distributions are narrower than the prior, with slightly larger mean values.

For each dynamic model in the ensembles, we evaluate gross physical quantities describing the rupture process, such as the mean slip, equivalent rupture radius $R = \sqrt{S/\pi}$ ($S$ being the actual ruptured area), duration, average rupture speed, mean $D_c$, and average stress drop, radiated energy $E_r$, average fracture energy density (or fracture energy $G$ over unit area), radiation efficiency defined as $\eta = E_r/(E_r + G)$, and radiated-energy-to-moment ratio $E_r/M_0$. Figure 6 displays their marginal posterior distributions as histograms. All displayed parameters (perhaps except for mean $D_c$) are characterized by unimodal Gaussian-like probability density distributions. Average rupture duration, velocity, and stress drop are similar between the events despite the significant difference in their AS shapes (Fig. 1b). While the similarity of the rupture duration can be easily explained by the rupture

geometry (i.e., unilateral for Mw4.2 and circular for Mw4.5 event), the similarity in the average velocity and stress drop is clarified by the synthetic test described in Supplementary Text S1. It shows that the average rupture velocity, including its variability, is primarily governed by the dynamic rupture with heterogeneous parameters, and therefore remains unconstrained by the AS inversion. However, the average stress drop is well resolved according to our synthetic test, and thus, the agreement of the stress drop marginal PDFs between the two events can be considered an important result of the AS inversion, suggesting a constant stress drop scaling of these two events in the same tectonic setting of Central Italy.

Figure 6 further shows how the events clearly differ in rupture radius, mean slip, $E_r$, average fracture energy density, and mean $D_c$. This agrees with the constant stress-drop source scaling laws: the seismic moment differs approximately 4 times, and thus, the mean slip and rupture radius of the smaller event is consistently $\sqrt[3]{4} \doteq 1.6$ times smaller. Similarly, the radiated energy is roughly 4 times larger for the Mw4.5 event. Our results also agree with the empirical and theoretical findings that $D_c$ and fracture energy rate scale linearly with slip[46,47], i.e., 1.6 times in our case. Although these parameters follow the scaling

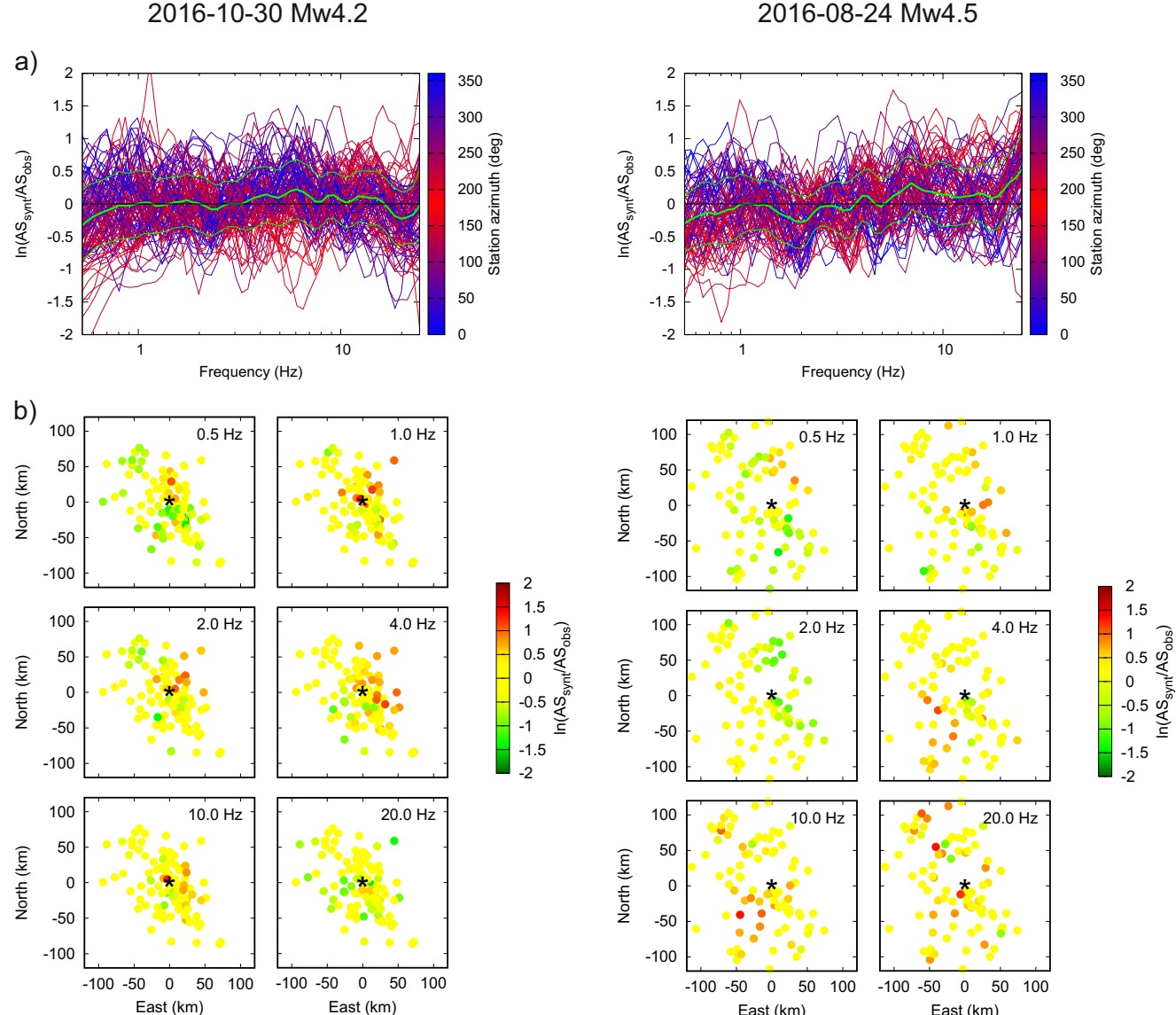

**Fig. 3 | Modeling spectral bias (difference between logarithms of the synthetic and empirical apparent spectra, AS) for the two analyzed events (columns).** **a** The bias for all receivers (lines color-coded by the station azimuth) as a function of frequency. Green thick and thin lines represent the average and standard deviations, respectively. **b** Spatial distribution of the bias plotted for six selected frequencies (see legend). Stars represent the respective event epicenters.

laws, the synthetic test (Supplementary Text S1) suggests that the AS constrain their values significantly compared to the assumed prior.

From the source scaling, it also follows that the scaled (non-dimensional) parameters, namely radiation efficiency and radiated-energy-to-moment ratio, seem magnitude-independent (Fig. 6). The inferred radiation efficiency of 0.15-0.25 suggests that approximately 75–85% of the available potential energy was dissipated in the rupture process. The radiation efficiency is relatively small compared to estimates for other earthquakes using various methods[2]. Nevertheless, similarly low values were also found for large crustal events using the same inversion technique applied to the waveforms and for various friction laws[23,24,26]. The inferred radiated-energy-to-moment ratio of $0.7\text{-}2 \times 10^{-5}$ is consistent with the range of estimates for other earthquakes[2].

### Beyond standard estimates based on corner frequency
In standard seismological practice, the source spectra are used to infer the corner frequency and estimate the stress drop from it. The standard omega-squared model of the source spectrum $\Omega(f)$ with corner

frequency $f_c$ reads

$$\Omega(f) = \frac{M_0}{1 + (f/f_c)^2} \tag{1}$$

We fit Eq. (1) to the synthetic MR spectra from our ensemble using L2 minimization in logarithms, treating scalar moment $M_0$ as known. Figure 7 shows the distributions of the estimated corner frequencies as histograms. Interestingly, the directive smaller event has generally larger corner frequencies (0.5–1.5 Hz) than the nondirective one (0.4–1.1 Hz) despite having similar durations (Fig. 6). As demonstrated by Liu et al.[10] and Gallovič and Valentová[9], the corner frequency is affected by the shape of the MR function, which is more complex in case of the directive event (Fig. 4). As shown in Supplementary Fig. S3, the fit is slightly worse for the nondirective Mw4.5 event but only above ~ 3 Hz (i.e., well above the corner frequency) because its spectral decay is faster than the omega-squared. Note that this corner frequency estimate represents an indirect approach because it relies on

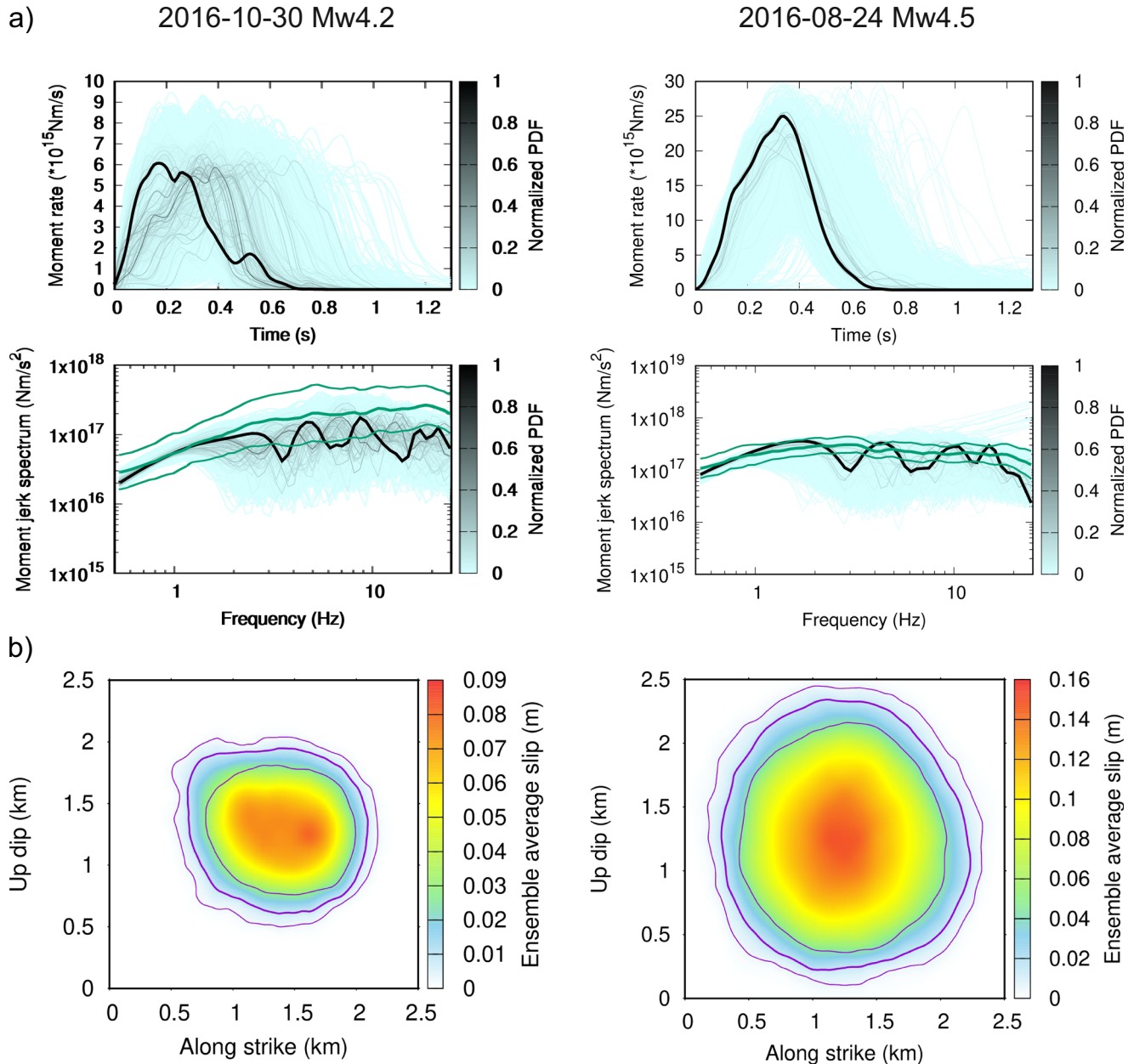

**Fig. 4 | Ensemble characteristics of the two inverted earthquakes (columns). a** Moment rates of all ensemble models color-coded by the normalized posterior probability density function (PDF) value (top) and the smoothed amplitude spectra of their second time derivatives (moment jerks, bottom). Green spectra correspond to the empirical station-averaged apparent spectra (AS), and standard deviations obtained from the Generalized Inversion Technique (same as in Fig. 1b). Thick black line corresponds to the maximum a-posteriori (MAP) model. Note the wide spread of the moment rates, which can be addressed to the missing phase information in the inverted data. **b** Ensemble-averaged slip distribution in color scale and contours of this average slip (thick line) with its ±1 sigma uncertainty (thin lines). Note the smooth character of the average slip distribution, contrary to the slip of the MAP model (Fig. 2).

fitting the MR spectra that can be obtained only by a finite-fault inversion.

We also fit the empirical spectrum of Eq. (1) directly to the apparent source spectra from GIT shown in Fig. 1b to obtain a standard seismological estimate. We calculate the L2 norm over all stations, effectively averaging over all azimuths and distances, and depict them by arrows in Fig. 7. While the resulting corner frequency of the non-directive Mw4.5 event falls within the values inferred by the dynamic inversion, the direct estimate from the GIT spectra is significantly overestimated for the directive Mw4.2 event. This is understandable when looking at the great (mainly azimuthal) variability of the GIT spectra shapes, here interpreted as due to the directivity effect.

The corner frequency is generally related to spectral stress drop $\Delta\sigma$ as

$$\Delta\sigma = \frac{7}{16}M_0\left(\frac{f_c}{k\beta}\right)^3 \tag{2}$$

where $k$ is a model-dependent parameter. Typically used are the Brune[7] model with $k = 0.37$ and the Madariaga[8] model with $k = 0.21$.

Histograms in Fig. 7 show the Brune and Madariaga stress drops calculated using Eq. (2) from the indirectly estimated corner frequencies (i.e., from the ensemble MR spectra). Note that both events are characterized by similar stress drops, as was

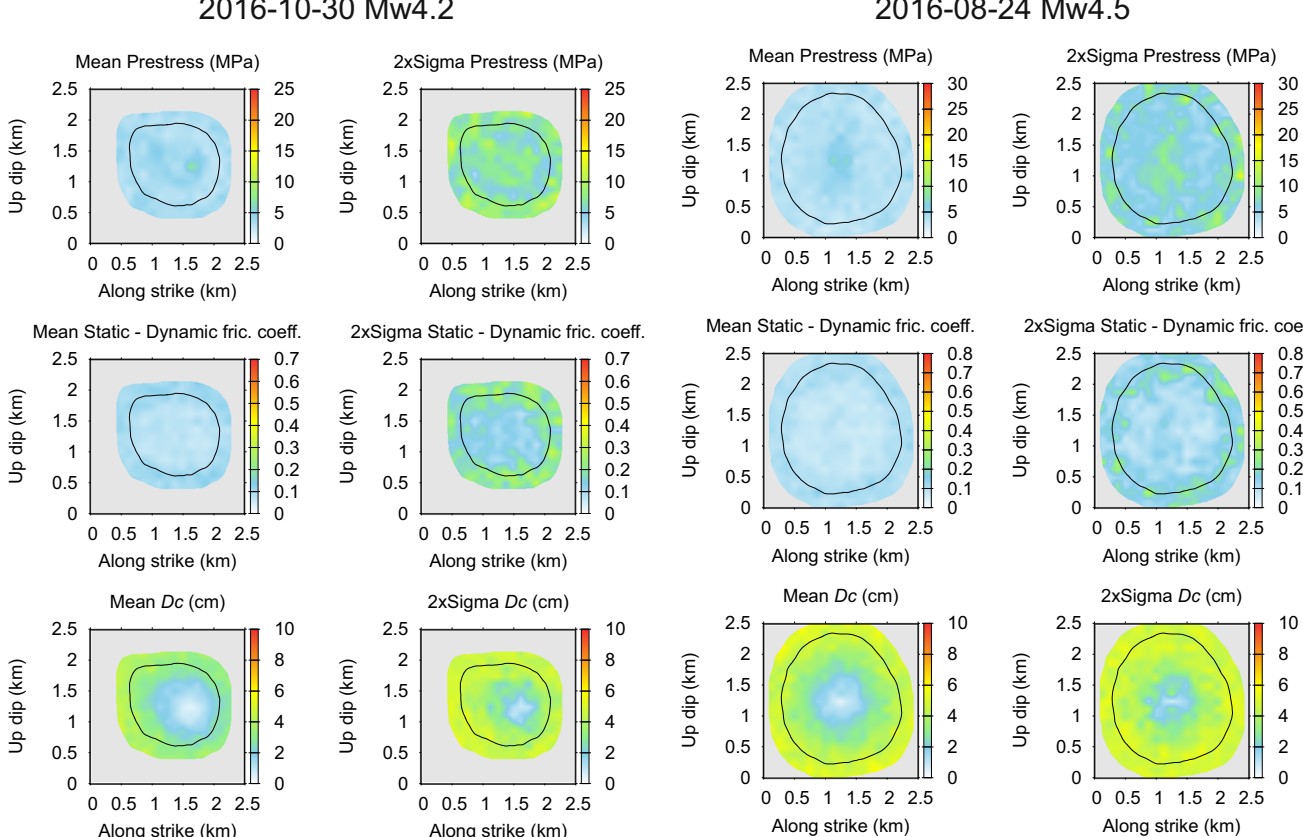

**Fig. 5 | Ensemble averages of the dynamic rupture parameters of the two studied events (left, right) along the fault and their ± 2 sigma uncertainty.** The unconstrained part of the model is masked by gray. The color scales are intentionally the same as for the maximum a-posteriori (MAP) model in Fig. 2 to enhance the smoothness of the ensemble-averaged spatial parameter distributions. Note also the significant variability around the mean model.

also observed for the stress drop calculated from the inferred dynamic models (Fig. 6). Nevertheless, the Madariaga stress drops are closer to the true stress drops derived from the dynamic models (Fig. 6) compared to those estimated by the Brune model. This supports the suitability of the Madariaga model, as it is based on dynamic simulations and aligns well with the results of Kaneko and Shearer[19] for a symmetric circular dynamic rupture using stacked apparent spectra (over station azimuths) and fitted by the omega-squared function.

We also calculate the Madariaga stress drop using the corner frequencies inferred directly by fitting the GIT AS to the omega-square model (shown as arrows in Fig. 7). The stress drop is overestimated significantly for the directive Mw4.2 event, resulting from the over-estimated corner frequency. We attribute this to the directivity of the rupture model and the insufficient averaging over the focal sphere[19]. This suggests that the simple spectral stress drop estimates from averaged AS (provided physics-based models, such as Madariaga, are utilized) may be biased in practice, especially for events with strong azimuthal variability of the AS. Therefore, the practitioners should first examine the variability of the AS shapes as their large spread may indicate a directive event for which the standardly inferred source properties might be significantly biased (Fig. 4a). A potential improvement may be in such cases represented by fitting apparent (station-dependent) corner frequencies estimated from the AS using asymmetrical rupture models[20]. However, the most reliable approach, regardless of the event's directivity, is to derive directly the moment rate functions by, e.g., dynamic or kinematic finite-fault inversion[48]. Improved stress drops can also be obtained from rupture area estimation by the second moment approach[49–51] to avoid using corner

frequency and the underlying assumptions on the temporal rupture characteristics[9].

**Spectral properties of the dynamic rupture spatial parameters**
To characterize the spatial properties of the dynamic rupture parameters, we explore the power spectral density (PSD) of the perturbations with respect to the average distribution. We recall that the average distributions are relatively smooth (Fig. 5), in contrast to the heterogeneity of the individual models (e.g., Fig. 2).

Let us select one of the dynamic parameters, $g(x, y)$, where $x$ and $y$ are the fault coordinates. We define the perturbations as $d(x, y) = g(x, y) - E(g(x, y))$, where $E$ represents the expectation evaluated by averaging the dynamic parameter over the model ensemble. The PSD is defined as

$$P\left(k_x, k_y\right) = \frac{1}{S} E\left( \mathcal{F}\left[d(x,y)s(x,y)\right]^2 \right) \tag{3}$$

where $\mathcal{F}$ represents the Fourier transform to wavenumbers $k_x$ and $k_y$, and $S$ is the fault area. Spatial taper $s(x, y)$ is the slip distribution $\Delta u(x, y)$ normalized by the square root of its average power, $s(x,y) = \Delta u(x,y)/\sqrt{\int \Delta u(x,y)^2 dS/S}$, to analyze only the ruptured part of the fault. We perform the analysis for $g$ being the initial stress and strength excess (strength minus the initial stress). Since the slip-weakening distance is presumably a scale-dependent parameter that varies over an order even in our models, we consider $g = \log D_c$. This corresponds to the analysis of the relative perturbations with respect to the median of $D_c$.

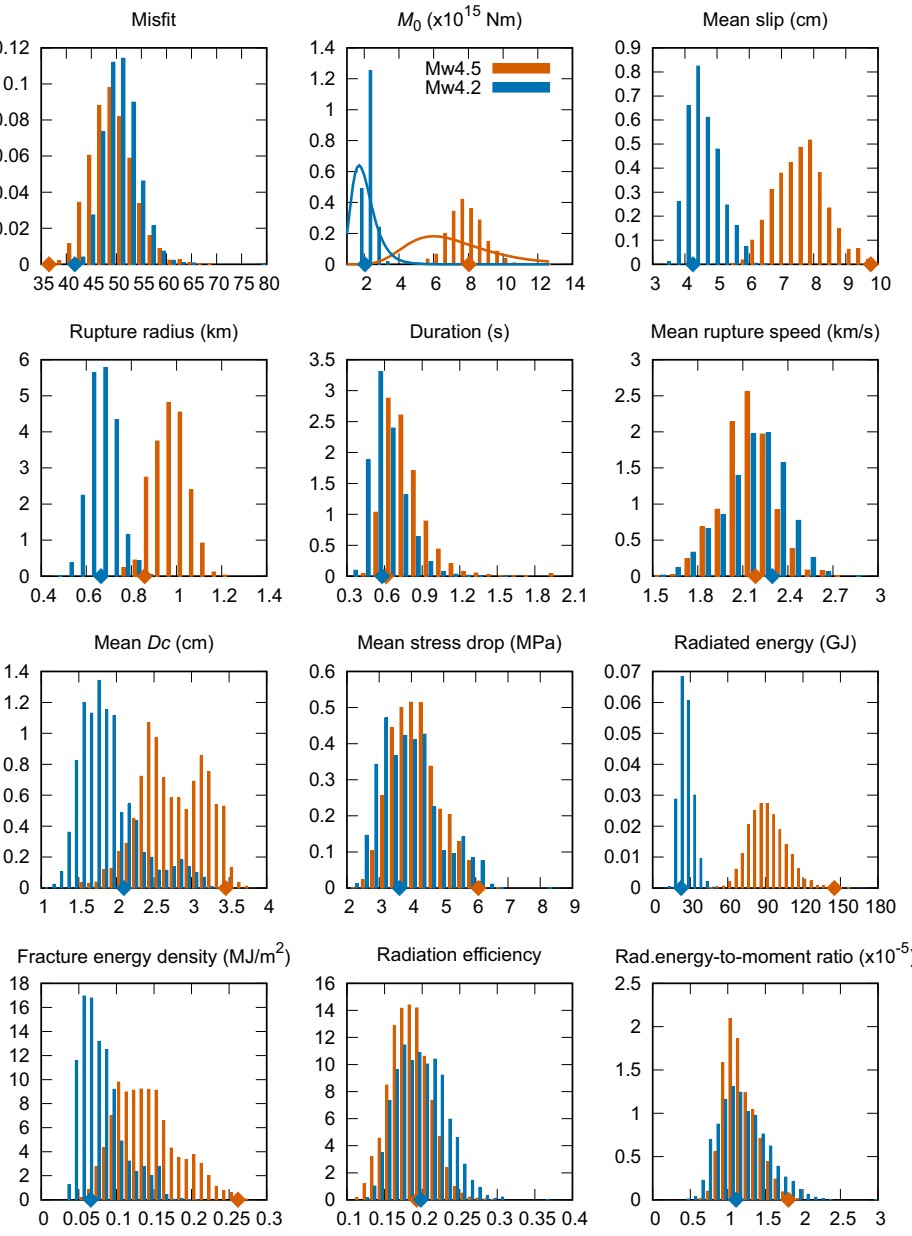

**Fig. 6 | Ensemble properties of various gross parameters (see legend) of the inverted rupture models for the two events (Mw4.5 in orange and Mw4.2 in blue), expressed as marginal posterior histograms normalized to unit** integrals. Diamonds show the parameters of the maximum a posteriori (MAP) models. In the seismic moment $M_0$ plot, the lines correspond to the prescribed prior distributions.

Figure 8 shows the resulting PSD estimates as a function of radial wavelength ($\lambda = k^{-1} = \left(k_x^2 + k_y^2\right)^{-1/2}$). The PSD exhibits $k^{-2}$ power-law spectral decay, corresponding to Hurst exponent $H = 0$, between approximately 200-600 m long wavelengths. The maximum 600 m long wavelength is close to the estimated radius of the ruptures. Thus, the correlation length of the heterogeneities is longer than or similar to the rupture size. The minimum 200 m long wavelength can be associated with the bilinear interpolation of the parameters used in the inversion. Indeed, the spatial discretization is 125 m, and thus, the faster power-law decay at shorter wavelengths can be attributed to the smoothing effect of the spatial interpolation.

The inferred $k^{-2}$ power-law decay of PSD corresponds to $k^{-1}$ decay of the amplitude spectrum. Such decay was considered for the initial stress by, e.g., Ripperger et al.[35] and Baumann and Dalguer[34] in their dynamic rupture simulations. They followed the theoretical derivation by

Andrews[52] based on the assumption of geometrical self-similarity of the ruptures. Smith and Heaton[53] estimated the falloff exponent of 1.3 for stress from the focal mechanisms of earthquakes in southern California.

The standard deviation of the perturbations (which, by definition, corresponds to the square root of the integral of PSD over wavenumbers) is specified in the respective panels of Fig. 8. The perturbations of the initial stress and strength excess are similar, 3-4 MPa. The inferred variability of $\log D_c$ (0.46–0.53) corresponds to the relative perturbation by 50–70% around the median value.

We further evaluate the coherence between the parameter perturbations $d_1$ and $d_2$ defined as

$$\gamma_{12} = \frac{E\left[\int d_1(x,y)d_2(x,y)s(x,y)dS\right]}{\sqrt{E\left[\int d_1^2(x,y)s(x,y)dS\right]}\sqrt{E\left[\int d_2^2(x,y)s(x,y)dS\right]}} \quad (4)$$

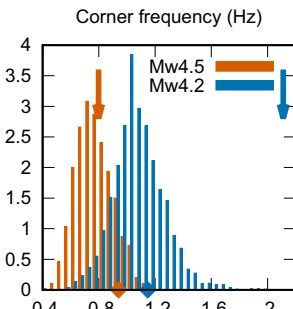
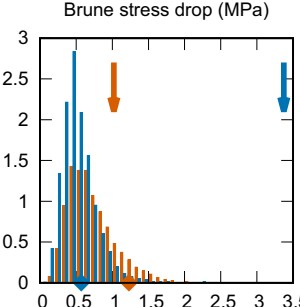
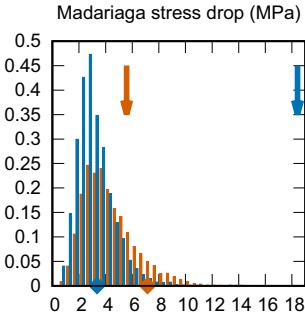

**Fig. 7 | Ensemble properties of the inverted source spectra for the two events (Mw4.5 in orange and Mw4.2 in blue).** Corner frequencies $f_c$ are obtained by fitting an omega-squared spectrum to the moment rate spectra of the individual models (i.e., indirectly through the finite-fault inversion result, see Fig. 4a). The Brune and Madariaga stress drops are calculated following Eq. (5) from the corner frequencies and scalar seismic moments, see text. Diamonds show estimates for the maximum a posteriori (MAP) models. Note the agreement between the stress drops of the dynamic models (Fig. 6) and the stress drop estimates using the Madariaga approach. Arrows denote respective results from the same spectral fitting applied directly to the observed station-averaged apparent spectra (AS). The overestimated values of $f_c$ and consequently stress drop for the directive Mw4.2 event likely result from averaging empirical AS over stations with limited coverage of the focal sphere, which biases their estimate (Fig. 1b).

The coherence between each two parameters from the analyzed group (prestress, strength excess, and $\log D_c$) is < 0.1. This suggests that the perturbations of the three dynamic rupture parameters are generally statistically independent. We also point out that the overall incoherence in the variations of the dynamic parameters does not imply complete independence because not all random combinations of the parameter perturbations would yield a self-sustaining rupture. Indeed, the coherence analysis is limited to identifying simple, approximately linear-like relationships and may fail to capture more complex, nonlinear dependencies governed by the dynamic model. Moreover, we cannot rule out possible coherence of source features at large scales, such as the correlation of the high stress drop and high fracture energy inferred from detailed dynamic rupture inversions of, e.g., the 2020 Mw 6.8 Elazığ earthquake by Gallovič et al.[24].

Let us point out that we use an idealized planar fault in the present application. In reality, rupture processes are affected by a range of physical phenomena not captured in this model, such as heat production, off-fault fracturing and plastic deformation, fault non-planarity, heterogeneous fault rheology, etc. As a result, the dynamic parameters inferred here should be regarded as effective representations of more complex underlying processes. As such, the inferred heterogeneities introduce sufficient complexity in rupture propagation to reproduce the observed AS. Direct association of our results with real physical fault zone complexities is beyond the scope of the present paper, but it can be the aim of further research.

## Discussion

We have developed and applied a dynamic rupture inversion with heterogeneous parameters to two small Mw- 4 earthquakes using apparent source spectra derived by GIT across a wide frequency range (0.5–25 Hz). It surpasses standard finite-fault inversions, which typically rely only on low-frequency (<1 Hz) waveforms due to the limited knowledge of the Earth's structure. Consequently, our approach enables a physically consistent dynamic inversion of small earthquakes, capturing high-frequency radiation and rupture heterogeneity that would otherwise remain unresolved. On the other hand, it comes at the cost of excluding the spectral phase information, which limits the ability to resolve the details of the rupture temporal evolution. Our Bayesian inverse formulation provides insight into which parameters are more robustly constrained by the data. Whereas the exact positions of rupture heterogeneities remain unresolved, the models capture the source complexity through randomly distributed heterogeneities along the fault. Importantly, the use of dynamic formulation ensures physical consistency, providing reliable constraints on the overall rupture process. Indeed, purely kinematic inversion of

AS would be highly unconstrained with a much larger variety of kinematic models fitting the AS data equally well.

Our results show that mainly the gross parameters (averaged over the rupture) can be inferred with reasonable uncertainty, such as the spatial extent of the rupture, mean slip, rupture speed, stress drop, radiated energy, radiation efficiency, and power spectral densities of spatial heterogeneity of dynamic parameters. Broadband spectrum inversion can also remove the potential bias between stress drop and rupture complexity, affecting the high-frequency source spectra with distinct azimuthal dependence. The Bayesian approach, in turn, guarantees that any irresolvable trade-offs are rendered in the parameter uncertainty.

Thanks to the consistent treatment of the Mw4.2 and Mw4.5 earthquakes, we can compare their source parameters. For example, the events are characterized by the same (within uncertainty) stress drop, rupture speed, radiated-energy-to-moment ratio, and radiation efficiency despite their different magnitudes. Other parameters, such as slip, spatial extent of the rupture, average $D_c$, etc., seem to follow approximately the constant stress drop scaling. Note that although such scaling is generally expected, it is demonstrated here by dynamic rupture inversion for heterogeneous spatial distributions of the source parameters of such small earthquakes. To properly test whether these properties hold for Central Italian earthquakes in general, the analysis developed in this study should be applied to a comprehensive set of events, which is an objective of a future study.

Whereas classical dynamic models considering smooth ruptures with sudden arrest at strong barriers can explain the observed approximately omega-squared apparent source spectral falloff, here we show that the same behavior can also be explained by heterogeneous ruptures with fractal properties. Indeed, such models offer a more realistic description of the inherently complex faulting process governed by a multitude of physical processes. Our estimated power spectral densities of the spatial heterogeneities of all dynamic parameters exhibit $k^{-2}$ power-law spectral decay. The analysis also suggests that the heterogeneity of the prestress, strength excess, and $D_c$ perturbations are uncorrelated, i.e., they vary effectively independently or with some more complex dependence undisclosable by the coherency analysis. Future studies are needed to test whether the fractal variations in prestress, strength excess, and $D_c$, inferred for our two events with distinct AS shapes, apply to other events and magnitudes.

Our physics-based inversion of rupture parameters is a promising way to surpass the simplifying assumptions in standard seismological approaches based on, e.g., Brune[7] or Madariaga[8] spectral source models to estimate stress drop. For example, the systematic azimuthal variability of the empirical apparent source spectra of one of the events can be explained only by an asymmetric rupture propagation

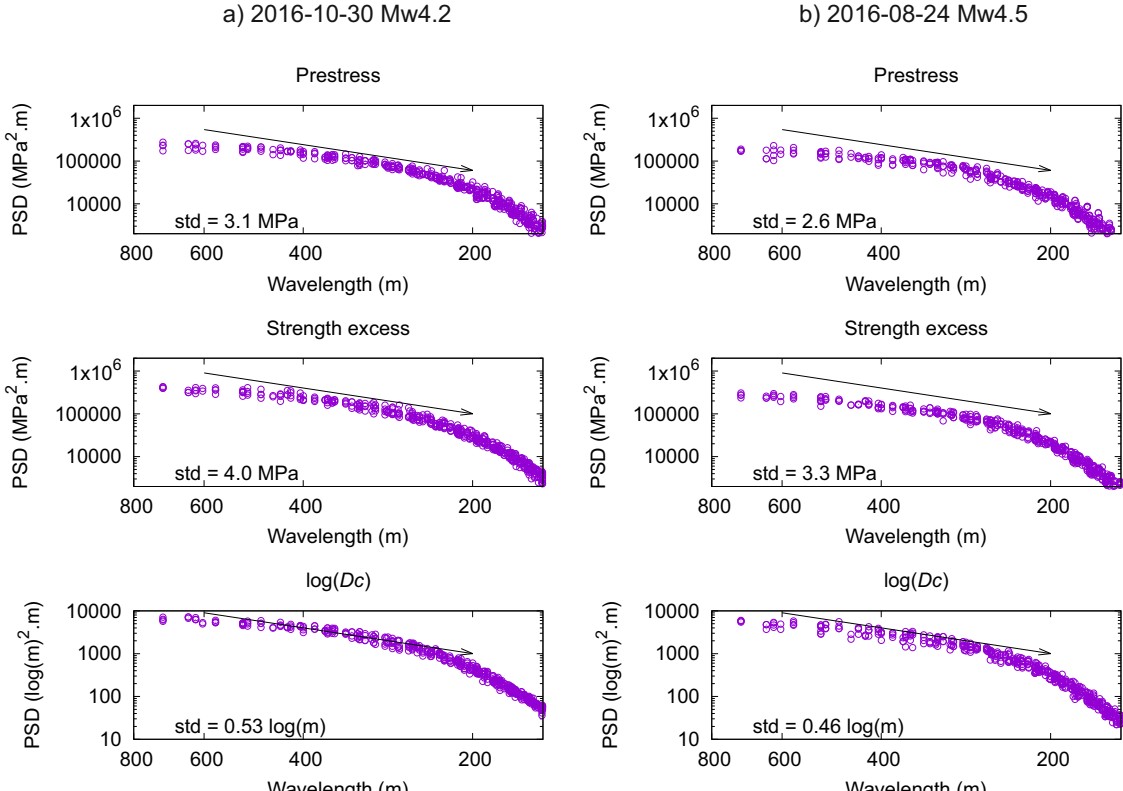

**Fig. 8 | Analysis of the statistical spectral properties of the spatial variability of dynamic rupture parameters across the ensemble models.** Panels show the power spectral density (PSD) of the perturbations estimated from the posterior ensembles of the dynamic rupture parameters with respect to their averages (for prestress see Fig. 5) for the two analyzed earthquakes (columns), plotted as a function of wavelength (the inverse of wavenumber $k$), displayed from long to short wavelengths (i.e., decreasing from left to right). The arrow indicates $k^{-2}$ power-law spectral decay. The *std* value in each plot is the standard deviation of the perturbations (i.e., the square roots of their power).

with a well-constrained degree of source complexity. We have shown that for such directive events, using the standard source models leads to biased estimates of the source parameters and should thus be avoided in present routine seismological studies.

Advanced studies of small earthquakes promise to unravel the so-far elusive small-scale characteristics of earthquake ruptures, mainly when applied to a population of events. Such research can improve the determination of not only the physical parameters of small earthquakes but also the small-scale characteristics of earthquake ruptures. The latter could lead to improved broadband modeling of earthquake scenarios for seismic hazard analysis. This study represents a data-driven look into rupture properties down to very short scales.

## Methods
### GIT derived apparent spectra
We utilize AS obtained by the so-called non-parametric Generalized Inversion Technique, GIT[39–41,54], a method used to isolate frequency-dependent source, path, and site functions from seismic recordings. The geometric mean of the horizontal components of the Fourier Amplitude Spectra (FAS) of the dominant S-wave in the acceleration time window are smoothed by the Konno and Ohmachi[55] algorithm. Then the GIT inversion decomposes the smoothed FAS $A_{ij}$ at station $j$ for event $i$ at source-station distance $r_{ij}$ into source, path, and site components as

$$\log A_{ij}(f, r_{ij}) = \log \widetilde{S}_i(f) + \log P_{ij}(f, r_{ij}) + \log G_j(f) + \delta W_{ij}(f) \quad (5)$$

where $\widetilde{S}_i$ corresponds to the source term, $P_{ij}$ is the attenuation function comprising the geometrical spreading and frequency-dependent

attenuation, $G_j$ is the site response, and $\delta W_{ij}$ is the remaining residual term from the GIT spectral fitting. $P_{ij}$ is considered 1 at distance $r_{ij} = r_{ref} = 10$ km, which is also the minimum source-station distance permitted in the dataset. One or more stations are carefully selected as reference rock stations[56], for which a flat and non-amplified site response is assumed (i.e., $G_j = 1$). The site response functions at other stations are thus derived relatively to this reference set.

The GIT decomposition was performed over many crustal events and stations in Central Italy[42] (see also Data availability), considering amplitude spectra at 56 frequencies equally spaced on a logarithmic scale in the 0.5-25 Hz range. The solution of the system described by Eq. (5) was calculated following a non-parametric inversion scheme, i.e., no specific functional forms related to the source and propagation terms were prescribed. Therefore, the non-parametric inversion scheme permits capturing possible complexities not only in the path and site effect terms but also in the earthquake source term.

Here, we utilize the GIT apparent source term defined as $\log S_{ij} = \log \widetilde{S}_i(f) + \delta W_{ij}$. Thus, $S_{ij}$ corresponds to a station-specific S-wave spectrum recorded at a reference rock station located at the reference distance $r_{ref}$. To obtain the apparent source spectrum $\Omega_{ij}^{\mathrm{GIT}}(f)$, one must correct for the residual radiation, wave propagation effects in a half-space from the source to the station at reference distance, and for the attenuation of the shallow layers above the half-space. We assume

$$S_{ij}(f) = \Omega_{ij}^{\mathrm{GIT}}(f) \frac{F_{ij}^S}{4\pi\rho\beta^3 r_{ref}} \widetilde{G}_j(f) \quad (6)$$

where $\rho$=3.1 gcm$^{-3}$, $\beta$=3.5 kms$^{-1}$ are the density and shear wave speed at the source, respectively. $F_{ij}^S$ is the radiation pattern considering the same combination of horizontal components as in GIT (i.e., geometric mean). Regarding the S-wave radiation pattern, we assume that wave scattering due to the random crustal heterogeneities disturbs the radiation pattern. We substitute this effect by calculating an average radiation pattern $\overline{F_{ij}^S}$ over random perturbations of the source mechanisms by ±30° in strike, dip, and rake. Term $\widetilde{G}_j(f)$ accounts for the source-to-surface amplification of the reference rock site. In general, the rock sites experience site amplification with respect to the homogeneous full space due to the free surface effect and the impedance contrast between the shallow S-wave velocity-density profile and the velocity-density at the source region. We assume $\widetilde{G}_j$ comprises the free surface effect (factor of 2), Boore and Joyner[57]'s generic rock site amplification, and shallow attenuation in the form of the high-frequency kappa effect with $\kappa$=0.015 s corresponding to rock site conditions in Central Italy[42,56]. We note that overall, $\widetilde{G}_j(f)$ varies only slightly between 3.4 at 0.5 Hz and 2.6 at 25 Hz, with a maximum of 4.7 at 4 Hz (see Supplementary Fig. S4). After correcting $S_{ij}$ for the terms in Eq. (6), the resulting $\Omega_{ij}^{GIT}(f)$ represents the apparent spectra (AS) that we use as data in the dynamic inversion.

We point out that the AS includes the term $\delta W_{ij}$ from GIT, which encompasses residual contributions from source, path and site-effect components. Nevertheless, the source radiation exhibits consistent, systematic spatial patterns across stations, producing a coherent signal within the AS. Therefore, when using a large number (~100 in our study) of stations in the dynamic rupture inversion, site and path residual effects primarily appear as unexplained posterior data errors and have only a minor impact on the inferred source models (see Supplementary Text S2).

We utilize the GIT-derived AS of two example events that are part of a high-quality catalog related to the latest major seismic sequence in the Central Italy area, which was densely recorded within 10–120 km and with a magnitude range of 3.2–4.5. The extensive collection was made possible thanks to the contributions from the stations of the temporary networks installed by the Italian Department of Civil Protection (DPC), as well as by several academic and research institutions, including the Istituto Nazionale di Geofisica e Vulcanologia (INGV); for more details, see Morasca et al.[42], Colavitti et al.[14] and Spallarossa et al.[58].

The non-parametric GIT decomposition is implemented using the open-source object-oriented Python package GITpy[59]. The AS are retrieved at ~100 stations (Fig. 1a). Hypocentral parameters are adopted from the Engineering Strong Motion database (see Data Availability) and listed in Table 1. The centroid moment tensors were calculated using BayesISOLA[60], and the fault planes were adopted from the resulting mechanisms. Of the two focal planes, we select the steeper ones that correspond to the orientation of the main tectonic structures in Central Italy. Nevertheless, we tried to calculate the AS for both planes, and the corresponding AS were almost the same, with a variance reduction of 0.999. This suggests that our method cannot reliably discriminate the fault plane for these two events. For more discussion, see also Supplementary Text S3.

## Dynamic rupture simulation and synthetic AS calculation

As a numerical solver, we utilize open-source code FD3D_TSN designed to simulate 3D dynamic earthquake ruptures[61]. It employs a high-accuracy, 4th-order finite-difference (FD) method on a Cartesian grid to solve the wave equation in a layered Earth model containing a vertical fault located on one face of the computational domain. The code implements a traction-at-split-node method to model fault friction and utilizes perfectly-matched-layers absorbing boundaries to minimize reflections from the artificial domain margins. FD3D_TSN employs OpenACC directives of the Nvidia Fortran compiler for GPU acceleration for improved computational efficiency, achieving significant

**Table 1 | Model and computational parameters considered in the dynamic inversion of the 2016-10-30 Mw4.2 and 2016-08-24 Mw4.5 events in Central Italy**

| Parameter | Mw4.2 | Mw4.5 |
|---|---|---|
| **General** | | |
| Seismic moment | 1.7e15 Nm | 6.0e15 Nm |
| Hypocenter location (latitude, longitude) | 42.8482°,13.0601° | 42.6171°,13.2905° |
| Hypocenter depth | 10 km | 6 km |
| Fault mechanism (strike, dip, rake) | 336°, 80°,− 90° | 132°, 47°, − 90° |
| Fault dimensions (length x width) | 2.5 km x 2.5 km | 2.5 km x 2.5 km |
| Normal stress (constant) | 100 MPa | |
| Cohesion | 0.5 MPa | |
| Dynamic friction | 0.4 | |
| P-wave and S-wave velocity, density | 6 kms$^{-1}$, 3.5 kms$^{-1}$, 2700 kgm$^{-3}$ | |
| **FD3D** | | |
| Spatial grid discretization | 10 m | |
| FD half-domain grid size (along strike x normal x along dip) | 250 × 125 x 250 | |
| Maximum duration of slip-rate functions | 2 s | |
| Time step | 0.4 ms | |
| **Apparent moment rate synthesis** | | |
| Spatial fault discretization | 10 m × 10 m | |
| Time sampling | 0.01 s | |
| Frequency range | 0.5 – 25 Hz | |
| **Model parameterization and prior parameter ranges (homogeneous along the fault)** | | |
| Control point grid (along-strike x along-dip) | 21 × 21 | |
| Initial stress prior | 0 – 100 MPa | |
| Static-to-dynamic friction coefficient drop prior | 0 – 1 | |
| Characteristic slip-weakening distance prior | 0.001 – 0.1 m | |
| Nucleation position prior (along-strike, up-dip) | 1.70 km, 1.25 km | 1.25 km, 1.25 km |
| Nucleation area prior (radius) | 0.2 km | |
| Maximum mean overstress in the nucleation area | 1 MPa | |

Hypocenter location was adopted from the ESM database.

(~10x) speedups compared to CPU execution. The code has been validated against established SCEC benchmarks[62], ensuring the code's accuracy in simulating ruptures with heterogeneous material properties. See more details in Premus et al.[61].

We assume slip-weakening friction law with spatially variable dynamic parameters. Namely, we invert for prestress above the dynamic friction, characteristic slip-weakening distance $D_c$, and so-called friction drop (difference between the static and dynamic friction), assuming a constant dynamic friction coefficient. The parameters are defined on a grid of control points 125 m apart, from which they are bilinearly interpolated onto the finer FD grid. The grid size was chosen so that the reliably resolved maximal simulated frequency corresponds to the maximum frequency of the employed data, 25 Hz (compare with, e.g., Valentová et al.[63]). For more details, see e.-g., Gallovič et al.[23,64]. Normal stress is assumed to be 100 MPa and constant over the fault plane. We prescribe a weak nucleation with a radius of 0.2 km and a maximum mean overstress of 1 MPa (Table 1) to avoid a strong effect on the synthetic AS at high frequencies.

Since the inverted events are small and relatively deep, we consider a dynamic rupture on a planar fault $2.5 × 2.5$ km$^2$ embedded in a

homogeneous space (see Table 1). The FD simulations are challenged by the maximum considered frequency of 25 Hz. As listed in Table 1, the discretization of the simulation domain for the FD simulations is very dense (10 m) with dense temporal sampling (0.4 ms). Still, the dynamic rupture simulation takes less than 10 seconds at Nvidia GTX 4070 graphics card. We tested the numerical convergence of the dynamic rupture simulation by decreasing the spatial and temporal grid size to half (5 m, 0.2 ms) for the MAP models. The models differ by 2–4% in their AS misfit, which is negligible considering the frequency range up to 25 Hz.

After simulating the rupture for a given distribution of dynamic parameters (event $i$), slip rates $\Delta \dot{u}_i(\boldsymbol{\xi}, t)$ are shifted in time following the S-wave propagation in a homogeneous space and integrated over the fault plane $S$ to obtain apparent source time functions (ASTF) for station $j$,

$$\Psi_{ij}^{synt}\left(t, \mathbf{x}_j\right) = \mu \int_S \Delta \dot{u}_i\left(\boldsymbol{\xi}, t - \frac{|\mathbf{x}_j - \boldsymbol{\xi}|}{\beta}\right) d\boldsymbol{\xi} \tag{7}$$

where $\mu$ is the shear modulus, $\mathbf{x}_j$ is the $j$-th receiver location and $\boldsymbol{\xi}$ denotes the position of the slip rate on the fault. The synthetic AS is then obtained by calculating the Fourier amplitude spectrum of the second-time derivative of Eq. (7), $\Omega_{ij}^{synt}(f) = \left|\mathcal{F}\left(\ddot{\Psi}_{ij}^{synt}\right)\right|$, where $\mathcal{F}$ is the Fourier transform and $|.|$ is the absolute value. As a final step, we apply the same smoothing to the synthetic AS as used in the empirical AS and evaluate it at the identical set of 56 frequencies in the same range of 0.5–25 Hz.

## Model parameterization and Bayesian inversion

We use Bayesian formalism for the inverse problem, where the solution is defined as a posterior probability density function (PDF) combining likelihood and prior PDFs[64]. The misfit defining the likelihood function is assumed to be the L2 norm between the natural logarithms of empirical and synthetic AS values, normalized by the square of data error equal to 4; for more information on the error estimate, see Supplementary Text S2. The prior is assumed to comprise the following constraints: (i) Seismic scalar moment $M_0$ being equal to the value from centroid moment tensor inversion (see Table 1) with $M_w$ uncertainty $\pm 0.05$. This can also be interpreted as adding a zero-frequency amplitude to the data. (ii) To ensure a weak nucleation, we assume that the average overstress cannot exceed 1 MPa within a 200 m radius around the hypocenter (Table 1). (iii) The friction drop and prestress are nonnegative and limited, and iv) $D_c$ must be larger than 1 mm for the sake of the numerical precision.

For the posterior sampling, we utilize the parallel tempering method[65], based on the Markov chain Monte Carlo (MCMC) approach. In this method, the MCMC exploration is performed in parallel at various temperatures, with the possibility of swapping temperatures based on an appropriately adapted Metropolis-Hastings criterion, allowing the exploration of larger areas of parameter space.

The parallel inversion is started from a suite of simple/smooth starting models with elliptical or circular rupture with various rupture speeds and stress drops. First, we created a dynamic rupture model with constant prestress and strength within an ellipse. Outside this region, the prestress is assumed to be half and the strength twice as high than on the inside, ensuring a gradual arrest of the rupture. Slip-weakening distance $D_c$ is considered to increase linearly from the hypocenter at a prescribed rate. This ensures that the nucleation is weak, almost negligible in the ASTF and AS, and the rupture propagates at a constant speed (see below).

We introduce variability into the initial smooth models by multiplying the prestress by a random factor drawn from a Gaussian distribution with a standard deviation of 2. To keep the rupture of these

perturbed models propagating and the seismic moment approximately unchanged, we adjust the dimensions of the source model and the $D_c$ rate from nucleation, following scaling relations derived below. We additionally change the $D_c$ rate randomly within $\pm10\%$ to introduce variability in the rupture speed.

Supplementary Fig. S2 shows two representatives of smooth models of the directive event, one with symmetric and one with asymmetric rupture propagation. The rupture gradually arrests at the edges of the elliptical source region due to the prescribed weak barrier represented by decreased initial stress and increased strength as described above. Both models underestimate the observed AS at frequencies > ~5 Hz. Our symmetrical model does not capture the azimuthal dependence of the observed AS. Interestingly, this holds even for the asymmetrical model, confirming the sensitivity of the synthetic AS to the rupture evolution.

Note that prescribing a stronger barrier with a fast rupture propagation would result in omega-squared spectra due to the radiation from the sudden stop of the rupture in some directions, as shown in the classical models of Sato and Hirasawa[66] and Madariaga[8] and extensively studied by Kaneko and Shearer[19]. Supplementary Text S4 describes the performance of several simulated smooth models surrounded by the strong barrier, suggesting that the rupture must be relatively fast to generate enough high-frequency radiation at its arrest, unlike the heterogeneous models with lower rupture speeds. Although such parameterization warrants further testing against real observations, it would require additional data, e.g., data sensitive to rupture velocity, which is beyond the scope of this work. Here, we employ parameterization with generally heterogeneous dynamic parameters, which naturally incorporates multiple deceleration and acceleration episodes at various scales and is thus considered more realistic when modeling real data (see Results).

To test the resolving power of the Bayesian dynamic source inversion of the amplitude AS, we designed a synthetic test based on the directive Mw4.2 event (Supplementary Text S1). The target model used to generate AS data for the same set of stations is a directive model with additional random heterogeneities in dynamic parameters for increased complexity. The MCMC inversion is run with the same settings as for the real event. For a detailed description, see Supplementary Text S1.

For each of the real Mw~4 earthquakes, we performed the parallel tempering sampling of the posterior PDF in 6 parallel jobs on 3 GPU Nvidia RTX 4070 cards, each job with 8 chains with 6 random temperatures $T$ between 1 and 100 and 2 temperatures $T = 1$. Each chain started from a randomly altered initial model. The entire inversion ran for > 30 days, during which each chain performed more than 5000 steps, i.e., about 250 k models were visited. The posterior analysis is conducted from samples at $T = 1$ after discarding 20% of the initial steps (burn-in phase) and keeping only every 10th accepted model to manage the storage size of the output models.

## Generation of initial elliptical models

To start the MCMC random walk, we use a set of various randomized smooth initial models. We note that most of the model space is filled with models that fail to even nucleate rupture. Our strategy is to generate an ensemble of reasonable models that are characterized by a priori known (approximately) seismic moment $M_0$.

We first create a dynamic rupture model that fits the assumed seismic moment and the low-frequency range of the observed apparent source spectra. Its rupture characteristics stem from the following equations. A simple circular crack model predicts the dependence between the seismic moment, constant stress drop $\Delta\sigma$, and radius $a$,

$$M_0 = \frac{16}{7} \Delta\sigma a^3 \tag{8}$$

We assume standard values of stress drop (~3 MPa for Central Italy, e.g., Pacor et al.[16]), and use Eq. (8) to infer the corresponding

radius of the source. The nucleation point is set either in the center if no directivity is observed in the empirical spectra or shifted by trial-and-error to either side of the rupture. For characteristic slip-weakening distance $D_c$, we assume it increases with hypocentral distance $\rho$ (in km) from minimum value $D_c^0$ at rate $D_c^{\text{rate}}$,

$$D_c = D_c^0 + D_c^{\text{rate}}\rho \tag{9}$$

This way, the nucleation can be kept small, and the rupture propagates at approximately constant speed. In the next section, we derive the scaling relation of $D_c^{\text{rate}}$ with $\Delta\sigma$ and strength drop $\Delta\tau_s$. It reads

$$D_c^{\text{rate}} = \frac{\pi}{\mu}\tilde{c}(v)\frac{\Delta\sigma^2}{\Delta\tau_s} \tag{10}$$

where $\tilde{c}(v)$ is a non-dimensional parameter between 0.1-1, depending on rupture velocity $v$, rupture mode and shape, etc. (see below). We found that in our case of having nucleation sizes of diameter ~ 0.1 km, a reasonable choice of $D_c^0$ is its value predicted by the $D_c^{\text{rate}}$ value at the edge of the nucleation patch, i.e., $D_c^0 = 0.1 D_c^{\text{rate}}$.

All the above equations can be used to define an initial rupture model. We then manually adjust and test these parameters to obtain a reasonable fit up to about twice the corner frequency. Note that resulting rupture models usually exhibit strong decay above their corner frequencies due to the smoothness of the dynamic parameters and gradual rupture arrest[33], see also Supplementary Fig. S2. This is later remedied by the inversion, permitting small-scale variations in the dynamic parameters.

After obtaining a reasonable single model, the next step is to create an ensemble of randomly varied models to initiate the MCMC inversion. We assume random variations of the above-found stress drop (e.g., by a factor of 2), and use Eq. (8) to rescale the size of the rupture model to keep $M_0$ constant. Similarly, $D_c^{\text{rate}}$ is scaled following Eq. (10), considering $v$ constant. Nevertheless, we then randomly perturb $D_c^{\text{rate}}$ by 10% to introduce variations in the rupture velocities. Note that we arbitrarily consider $\Delta\tau_s$ unperturbed, but its additional random perturbations can be assumed as well.

### Scaling of the distance-dependence of $D_c$

We assume an expanding self-similar anti-plane (mode III) 1D crack with constant speed $v$, for which Broberg[67] derived the stress intensity factor at the crack tip,

$$K_{III}(v) = \Delta\sigma\sqrt{\pi v t}\frac{\gamma_\beta(v)}{E\left(\gamma_\beta(v)\right)} \tag{11}$$

where $\Delta\sigma$ is the stress drop,

$$\gamma_\beta(v) = \sqrt{1 - \left(\frac{v}{\beta}\right)^2} \tag{12}$$

and $E$ is the complete elliptical integral of the second kind. For such a model, the dynamic energy release rate reads

$$G_{III}(v) = \frac{K_{III}^2(v)}{2\mu}\frac{1}{\gamma_\beta(v)} \tag{13}$$

Combining Eqs. (11) and (13) yields

$$
\begin{aligned}
G_{III}(v) &= \frac{\Delta\sigma^2\pi v t \gamma_\beta^2(v)}{2\mu E^2\left(\gamma_\beta(v)\right)}\frac{1}{\gamma_\beta(v)} = \frac{\Delta\sigma^2\pi v t}{2\mu}\frac{\gamma_\beta(v)}{E^2\left(\gamma_\beta(v)\right)} \\
&= \frac{\Delta\sigma^2\pi v t}{2\mu}c(v),\ \text{with}\ c(v) = \frac{\gamma_\beta(v)}{E^2\left(\gamma_\beta(v)\right)}
\end{aligned}
\tag{14}
$$

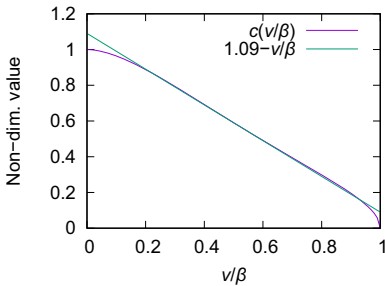

**Fig. 9 | Dependence of $c(v)$** defined in Eq. (14) on the ratio of rupture velocity $v$ and shear wave speed $\beta$ (magenta curve). Note that it is well approximated by the equation $1.09 - v/\beta$ (green curve) in the central part of the x-axis.

Assuming the rupture propagating over a fault with a given fracture energy rate $G$ will reach a steady-state velocity $v$ after extending to size $a = vt$, from Eq. (14) it follows:

$$G_{III}(v) = \frac{\Delta\sigma^2\pi c(v)}{2\mu}a \tag{15}$$

Considering a rupture governed by the slip-weakening friction, the fracture energy rate is

$$G = \frac{1}{2}D_c\Delta\tau_s \tag{16}$$

where $D_c$ is the slip-weakening distance and $\Delta\tau_s$ is the strength drop. Equating the fracture and dissipation energy rates (Eqs. 15 and 16) yields a condition for the increase of $D_c$ with the increase of crack size $a$,

$$D_c = \frac{\pi}{\mu}\frac{\Delta\sigma^2}{\Delta\tau_s}c(v)a \tag{17}$$

Figure 9 shows the dependence of $c$ on the ratio of $v/\beta$ and how it can be well approximated by a linear function for a wide range of reasonable rupture velocities.

Numerical experiments with circular and elliptical ruptures suggest that the $D_c$ rate must be 1.3-2.2 times smaller than predicted by Eq. (17), perhaps due to the transition from 2D to 3D rupture model. Therefore, $\tilde{c}$ in Eq. (10) is 1.3-2.2 times smaller than $c$. Equation (17) allows the expression of the rupture velocity as a function of the dynamic rupture parameters. For a reasonable rupture (subshear and sustained), min and max $v$ obtained assuming 2.2-1.3 times larger values than predicted by Eq. (17), respectively, should fall within 1-3 kms⁻¹.

## Data availability
The waveform data, locations and source metadata are provided through the website of the Engineering Strong-Motion (ESM[68,69]; https://esm-db.eu/) and the Italian ACcelerometric Archive (ITACA[70], https://itaca.mi.ingv.it/ under IDs 20161030_0000130 (Mw4.2) and 20160824_0000007 (Mw4.5). The FAS dataset is available from https://shake.mi.ingv.it/central-italy/. The data and results of the dynamic inversions presented in this work are available at Zenodo[71] (https://doi.org/10.5281/zenodo.17417423).

## Code availability
The GITpy code is accessible through https://gitlab.rm.ingv.it/inversion/gitpy. The moment tensor inversion package using BayesISOLA is available from https://github.com/vackar/BayesISOLA. All dynamic rupture simulations were performed using open-source code fd3d_tsn_pt (https://github.com/fgallovic/fd3d_tsn_pt) co-developed

by F.G., Ľ.V.K., and Jan Premus, see description in Premus et al.[61] and Gallovič et al.[23,64]. The source code of fd3d_tsn_pt is available at Zenodo[71] (https://doi.org/10.5281/zenodo.17417423).

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

## Acknowledgements

We are grateful to the INGV Milano colleagues (F. Pacor, L. Vitrano) for providing us with the source spectra from their GIT. This research was supported by the Johannes Amos Comenius Program (OP JAC), project No. CZ.02.01.01/00/22_008/0004605, Natural and anthropogenic georisks (F.G. and L.V.K.). We acknowledge financial support by The Ministry of Education, Youth and Sports of the Czech Republic through the program INTER-EXCELLENCE II (grant LUAUS23233) (F.G. and L.V.K.). This research was partially supported by the Istituto Nazionale di Geofisica e Vulcanologia (INGV) within the framework of the SECURE project - part of Pianeta Dinamico (Working Earth) - Geosciences for the Understanding of the Dynamics of the Earth and the Consequent Natural Risks (CUP code D53J19000170001), funded by the Italian Ministry of University and Research (MIUR) (S.S.).

## Author contributions

F.G. conceptualized and designed the Bayesian dynamic source inversion of the apparent source spectra, performed the calculations for the two events and the respective analyses. S.S. managed the acquisition and analysis of the GIT data. L.V.K. performed the synthetic tests and CMT inversions. All authors drafted the manuscript.

## Competing interests

The authors declare no competing interests.

## Additional information

**Supplementary information** The online version contains
supplementary material available at

František Gallovič.

**Peer review information** *Nature Communications* thanks Jan Dettmer,
and the other anonymous reviewer(s) for their contribution to the peer
review of this work. A peer review file is available.

