## [Transparent Peer Review file · Nature Communications]

Physics-based broadband characterization of weak earthquakes

Corresponding Author: Professor František Gallovič

Version 0:

Reviewer comments:

Reviewer #1

(Remarks to the Author)

Gallovic et al. employ a novel dynamic rupture inversion technique to investigate the rupture processes of two magnitude 4 earthquakes in central Italy. Their key conclusions are that small-scale (~100 m) heterogeneities in rupture propagation are essential to fit the high-frequency radiation and that fractally distributed dynamic parameters (prestress, friction drop, characteristic slip-weakening distance, etc.) are required to explain the data. While previous kinematic inversion studies and theoretical work have yielded similar insights, this study is the first to apply dynamic rupture inversion to reach the same conclusion, as the authors correctly claim. Although the results are potentially significant and the inversion's ability to accurately resolve only the average properties of source parameters is reasonable and useful for understanding the source processes of small earthquakes, I believe additional simulations and clarifications are necessary to support the main conclusions (as stated below).

Furthermore, this study analyzes only two small events, limiting the generalizability of the inferred source characteristics. As the authors acknowledge, further research is needed to determine whether these findings extend to other small earthquakes. However, I question the validity of the inferred fractalness, where pre-stress, strength excess, and D_c are all fractally distributed in the two analyzed events. While this might be valid for these specific cases, specific fractal distributions obtained in this study, particularly for the pre-stress and strength excess, cannot necessarily be extrapolated to either tiny or large earthquakes. If self-similarity holds, these fractal distributions would not be generalizable to other small earthquakes.

Major comments:

The authors state (Line 170) "We would like to emphasize that the smooth average dynamic model would be unable to fit the observed AS data (see Materials and Methods and cf. Fig. S2)." This is an important result and should be discussed in more detail. I recommend moving Fig. S2 into the main manuscript. More importantly, additional synthetic tests and alternative parameterizations of dynamic rupture models are needed to support this claim. Classical models by Sato and Hirasawa (1973) and Madariaga (1976), among others, incorporate a sudden rupture arrest, which results in an omega-squared high-frequency decay. However, Figure S2 suggests that the rupture gradually arrests at the edge of the source region, likely due to the assumptions made in setting up the "smooth" rupture model. For instance, Figure S2 indicates that, just outside the slip zone, the initial stress is only slightly smaller and the static minus dynamic friction coefficient is relatively large. This setup does not produce a sudden rupture arrest, which is typically required for high-frequency energy generation. If the authors were to implement a sudden arrest model, the results might differ significantly from those presented in Figures S2a and S2d. Given the importance of this conclusion, additional results and discussion are warranted.

The mean spectral bias between observed data and the MAP results (Figure 3a) indicates that low-frequency (<0.8 Hz) energy is systematically missing in the MAP model. Since the vertical axis is logarithmic, the discrepancy reaches approximately 30% at 0.5 Hz, which is substantial given that this is the most resolvable frequency range. Please provide an explanation for this discrepancy and justify the difference.

The authors state (Line 223) "We fit Eq. (1) using L2 minimization in logarithms to the synthetic MR spectra from our ensemble, considering scalar moment M_0 as known." Please include the fitting results, as they are crucial for assessing how

well Eq. (1) fits across a broad frequency range. Additionally, are there any biases between estimated f_c and the goodness of fit?

The authors state (Line 255) “This seems to contradict the findings of Yoshida (2019), who showed that stress drop estimates can be almost twice as low when the asymmetry of the rupture propagation is neglected.” More context is needed here, as the line of reasoning is unclear. First, what is meant by “the theoretical model of Yoshida (2019)” (Line 259)? Yoshida (2019) used the model of Dong and Papageorgiou (2003) rather than developing a new theoretical model. Furthermore, Kaneko and Shearer (2015) showed that their asymmetrical rupture model is set up to produce a longer source duration than the symmetrical model, reflecting the smaller k values in the asymmetrical model. Since the present study infers similar source durations for the two events, it is inappropriate to interpret the results using the Kaneko and Shearer (2015) models. Please provide more background before comparing the results to previous theoretical models.

The authors state (in Line 293) that the inferred k^{-2} decay of PSD corresponds to k^{-1} decay of the amplitude spectrum, and such a decay was considered for the initial stress in previous theoretical studies. However, the previous studies (e.g., Ripperger et al., 2007) assumed that stress heterogeneity is described by a power law decay of its wave number spectrum. I might be missing something here, but Figure 8 shows increasing amplitude with wavelength, and hence, the trend is opposite from those assumed in the previous studies.

Minor comments:

Line 19: The statement “Heterogeneity in the rupture propagation down to the smallest scales is necessary to fit the high-frequency radiation” could be misleading, as it does not consider the debate on whether high-frequency energy arises from elastic heterogeneities near the source rather than rupture propagation. Since this study does not account for elastic heterogeneities, the statement should be revised to avoid confusion.

Line 35: Yoshida (2024) should be corrected to Yoshida (2019).

Line 45: The phrase “can attain different than” is unclear. Please clarify.

Line 73: The claim that this study performs “an inversion of AS free of path and site effects” seems overstated. Complete removal of path and site effects is generally impossible, particularly due to elastic heterogeneities near the source (e.g., fault damage zones within a few hundred meters). Consider revising this statement.

Line 90: The phrase “described in Text S1” is inappropriate for the Introduction section, as it references results before they are presented. Move this sentence to the Results section.

Line 145: “due to the prescribed weak nucleation”. This is not explained anywhere.

Line 458: The authors should discuss numerical resolution in terms of cohesive zone sizes (e.g., Day et al., 2005), as is common in dynamic rupture simulation studies.

(Remarks on code availability)

Reviewer #2

(Remarks to the Author)
See attached PDF

(Remarks on code availability)

I have reviewed the code at a high level. It seems in good order and documented. However, I did not have time to actually run and test it. For a code at this complexity, that would take a time commitment of many hours.

Reviewer #3

(Remarks to the Author)

I read the paper “Physics-based broadband characterisation of weak earthquakes” by Gallic et al. with considerable interest. It is a fine piece of work and appears to be of high quality and completeness. It shows how spatial variations in earthquake amplitude spectra can be used to determine important details about the rupture properties of even relatively small earthquakes, in this case of magnitudes 4.2 and 4.5. It seems remarkable to see rupture directivity resolved for an earthquake as small as magnitude 4.2. Moreover it shows how failing to account for these details can sometimes lead to very significant bias in the conventional methods for estimating important parameters like stress drop. I think the paper is an important step forward in estimating earthquake source parameters.

However, while I myself found the paper very interesting I would have to say that I think it appeals mainly to a narrow audience of ground motion seismologists. If Nature is fine with accepting a paper that may not have broad appeal, then the paper can be published with only minor revision. If instead it is Nature’s goal to publish only papers with broad appeal, then this paper may not be such a good fit. In order to make it appeal to a wider audience I think it would require major revision, such as perhaps considering a much larger dataset - difficult, given that these two events each took 30 days of run time.

1) My main substantive comment is that in the section on “GIT derived apparent spectra”, some important assumptions are mentioned, such as the value of the high-frequency attenuation coefficient κ ($=0.015$, Line 411), and the use of random perturbations in the strike, dip, and rake to mimic random perturbations to crustal velocity structure. These all may be valid assumptions, but without further evidence I’m left wondering whether there could be trade-offs between some of the source properties and the κ value. Random crustal velocity perturbations will not only obscure the S-wave radiation pattern but can also produce frequency dependent apparent attenuation due to scattering. Could such effects be mapped into the source properties if they are not accounted for? In any case, I feel more discussion of these is warranted.

2) As with all Bayesian inversions, an important question is: Has it converged? With the sampling of 250k models over 30 days run time, I could well believe the solution has converged. However, some indication of how the authors assessed this, such as examining the character of the chain histories of likelihood values, should be discussed.

Line 519: Shouldn’t the total number of models visited by $5k \text{ steps} \times 8 \text{ chains} \times 6 \text{ jobs} = 240k$, not 250k?

Figure 1: It can be difficult to make out the stars that denote the epicentres. Could they be made bigger, or a different colour so they’d stand out more? The lines connecting the beachballs to the respective epicenters are also very thin.

Figures 6 & 7: These are a little confusing because the units of the x-axis at the bottom of the figure are actually given as part of the figure title, at the top of the figure. I recommend moving the units to the bottom of the figure, just below the x-axis, as has been done with other figures.

(Remarks on code availability)

Version 1:

Reviewer comments:

Reviewer #1

(Remarks to the Author)

I appreciate the authors' efforts to revise the manuscript and provide detailed responses to all comments. The study continues to offer a promising approach to dynamic rupture inversion for small earthquakes, and I acknowledge the added value of the new parametric study shown in the rebuttal. However, I am not satisfied with the revision in its current form.

My primary concern remains the same as in my initial review. I feel that the authors have not adequately addressed the first major comment of Reviewer #1 regarding the role of abrupt rupture arrest models. While I acknowledge that additional simulations were conducted and shared in the rebuttal letter, the results of this important analysis are not included in the revised manuscript or its Supplementary Information. Presenting such key results only in the rebuttal is inappropriate, as it prevents readers of the published paper from critically evaluating the comparison between abrupt arrest and heterogeneous rupture models.

The revised manuscript continues to emphasize the necessity of small-scale heterogeneity to fit the observed high-frequency spectra. This is a central conclusion of the paper and, if correct, would represent a significant advance. However, without properly documenting the performance of sudden arrest models—beyond the gradual-arrest smooth models shown in the SI—the claim that heterogeneity is required remains insufficiently supported. The authors themselves acknowledge that barrier models with fast rupture can produce omega-squared falloff and fit the observed spectra reasonably well under certain conditions. Yet this important nuance is hidden in the rebuttal and omitted from the actual manuscript.

If the barrier models performed significantly worse than the heterogeneous models, then the comparison should be documented and discussed explicitly in the paper. If the differences are more subtle—perhaps depending on trade-offs with rupture speed or stress drop—then that too should be part of the discussion. In either case, the reader must be given access to these results in order to independently evaluate the authors' interpretation.

I also note that the revised version makes several definitive statements implying that heterogeneity is required (e.g., Abstract, Lines 21–22: “In our models, the heterogeneity down to the smallest scales (~ 100 m) is required to fit the observed high-frequency radiation.”). In light of the alternative modeling presented in the rebuttal, such statements are too strong without presenting the full scope of tested models.

Therefore, I encourage the authors to revise the manuscript by:

- (i) Incorporating the abrupt arrest (barrier) model results into the manuscript or SI in a clearly labeled section, including key figures and discussion of misfits, parameter trade-offs, and spectral comparisons.
- (ii) Clarifying the framing of their conclusions regarding the necessity of heterogeneity in light of the alternative models.
- (iii) Moderating claims in the Abstract, Results, and Conclusions to reflect the fact that other models (with abrupt arrest) can also explain aspects of the high-frequency radiation, albeit with limitations or different trade-offs.

These additions would substantially improve the rigor, transparency, and impact of the paper. Until these revisions are made, I would not consider the current manuscript suitable for publication in Nature Communications.

Minor comment:

Line 62: The authors mention that “Yoshida (2019) demonstrated that asymmetrical rupture propagation may result in biased stress drop estimates if the directivity effect on the source spectra is neglected.” It could be helpful to also acknowledge that methods such as the second moment approach (e.g., McGuire et al., 2017; McGuire and Kaneko, 2018) offer an alternative pathway to improved stress drop estimates by better resolving source geometry and rupture propagation direction.

(Remarks on code availability)

Reviewer #2

(Remarks to the Author)

I would like to thank the authors for an exceptionally thorough response to my comments. I have no further concerns with this manuscript and recommend it be considered for publication. It's excellent work in my opinion.

(Remarks on code availability)

Version 2:

Reviewer comments:

Reviewer #1

(Remarks to the Author)

I appreciate the careful and constructive responses to the previous review comments. The authors have appropriately revised the manuscript text to ensure consistency with the presented results and have added a detailed description of the abrupt-rupture-arrest model simulations in the Supplementary Information (Text S4). Although they did not conduct a full inversion or quantitative comparison for abrupt-rupture-arrest models as I originally requested, the revised version now clearly articulates the underlying assumptions, limitations, and rationale of their analysis. (I still consider such an analysis to be an important next step for future work.) In any case, the central findings are now presented in a balanced and transparent manner. Given these revisions, I consider the manuscript suitable for publication in Nature Communications in its current form.

(Remarks on code availability)

Dear Editor and Reviewers,

We thank you for your time and efforts in carefully reading our manuscript and for the provided comments. We have addressed all of them as we describe below, and we believe this has improved our manuscript considerably.

Reviewer #1:

Galovic et al. employ a novel dynamic rupture inversion technique to investigate the rupture processes of two magnitude 4 earthquakes in central Italy. Their key conclusions are that small-scale (~100 m) heterogeneities in rupture propagation are essential to fit the high-frequency radiation and that fractally distributed dynamic parameters (prestress, friction drop, characteristic slip-weakening distance, etc.) are required to explain the data. While previous kinematic inversion studies and theoretical work have yielded similar insights, this study is the first to apply dynamic rupture inversion to reach the same conclusion, as the authors correctly claim. Although the results are potentially significant and the inversion's ability to accurately resolve only the average properties of source parameters is reasonable and useful for understanding the source processes of small earthquakes, I believe additional simulations and clarifications are necessary to support the main conclusions (as stated below).

Furthermore, this study analyzes only two small events, limiting the generalizability of the inferred source characteristics. As the authors acknowledge, further research is needed to determine whether these findings extend to other small earthquakes. However, I question the validity of the inferred fractalness, where pre-stress, strength excess, and D_c are all fractally distributed in the two analyzed events. While this might be valid for these specific cases, specific fractal distributions obtained in this study, particularly for the pre-stress and strength excess, cannot necessarily be extrapolated to either tiny or large earthquakes. If self-similarity holds, these fractal distributions would not be generalizable to other small earthquakes.

We agree with the reviewer that the inferred fractal characteristics of the dynamic parameters cannot be immediately generalized to other events and magnitudes without further studies. For now, we have found the fractal properties only of the two events, but more importantly, with significantly different shapes of their apparent spectra. We point out that the perturbations are described “on top of” the mean spatial spectra of dynamic parameters, and do not describe the mean spectra themselves. Based on the theoretical description of von Karman power spectrum and a few preliminary tests that we performed, the fractal properties can be rescaled both to larger and smaller events and generate various spectral behaviors. However, it warrants a devoted, in-depth study. We have added such a note in Conclusions: “Future studies are needed to test whether the fractal variations in prestress, strength excess, and D_c , inferred for our two events with distinct AS shapes, apply to other events and magnitudes.”

Major comments:

The authors state (Line 170) “We would like to emphasize that the smooth average dynamic model would be unable to fit the observed AS data (see Materials and Methods and cf. Fig. S2).” This is an important result and should be discussed in more detail. I recommend moving Fig. S2 into the main manuscript. More importantly, additional synthetic tests and alternative parameterizations of dynamic rupture models are needed to support this claim. Classical models by Sato and Hirasawa (1973) and Madariaga (1976), among others, incorporate a sudden rupture arrest, which results in an omega-squared high-frequency decay. However, Figure S2 suggests that the rupture gradually arrests at the edge of the source region, likely due to the assumptions made in setting up the “smooth” rupture model. For instance, Figure S2 indicates that, just outside the slip zone, the initial stress is only slightly smaller and the static minus dynamic friction coefficient is relatively large. This setup does not produce a sudden rupture arrest, which is typically required for high-frequency energy generation. If the authors were to implement a sudden arrest model, the results might differ significantly from those presented in Figures S2a and S2d. Given the importance of this conclusion, additional results and discussion are warranted.

We agree that the statement was too strong and thus confusing. Strong stopping phases would indeed result in omega-squared radiation in some directions, as analyzed in detail by, e.g., Kaneko and Shearer (2015). The topic of whether the rupture is necessarily heterogeneous is still open and very interesting, as it is difficult to extract from real observations.

Based on the reviewer’s suggestion, we performed additional synthetic tests, in particular a simple parametric study for the non-directive Mw4.5 event. In this case, the rupture geometry is simple—circular with nucleation in the center of the fault. The initial stress and strength were prescribed to be homogeneous inside the circle (except for the increased initial stress in the small nucleation patch), and the barrier around the circle was prescribed with zero initial stress and 15 times larger strength.

The parametric study was performed for two main parameters: stress drop and rupture velocity. The stress drop $\Delta\sigma$ (determined mainly by the initial stress) controls the size of the rupture with radius r based on the scaling relationship for a given seismic moment M_0 : $M_0 = \frac{16}{7}\Delta\sigma r^3$. We tested 3 values of $\Delta\sigma$: 2 MPa, 4 MPa, and 8 MPa. The second parameter, the rupture velocity, is determined by the prescribed Dc rate, describing the assumed linear increase of Dc from nucleation according to Eq. (17) in the main text. We assumed several values of rupture velocity ranging between 0.4 and 1.0 of β .

This simple setting results in models with almost constant stress drop and rupture velocity along the fault, i.e., 2-parameter space. For these barrier dynamic models, we compare the synthetic apparent spectra with the observed ones. The result of this simple parametric study with models color-coded by misfit (left) or spectral bias (right) is shown in this figure:

We note that the parameters shown in this figure (as well as the resulting moment) are based on the resulting dynamic simulation and thus slightly differ from the original (prescribed) value.

The best model has a misfit of ~ 52 . Based on this value, it might have been accepted in our ensemble if it had been present in the MCMC sampling. We note that the best model of this parametric test should not be compared with the best model of the Bayesian inversion, as in the former case, the parameters were not optimized. Nevertheless, based on this simple parametric study, we can infer some general characteristics of the barrier model with apparent spectra similar to the observations.

Properties of the best barrier model:

The best model has a stress drop of ~ 4.2 MPa; models with stress drop values of 2 and 8 MPa have higher misfit (>70). Thus, the obtained stress drop value lies within the marginal posterior distribution, as found by the Bayesian inversion (cf Fig. 6).

Contrary to the heterogeneous model, the barrier model requires a higher rupture speed (2.9 km/s, i.e., $V_r/\beta \sim 0.8$), likely to generate high-frequency radiation spectra comparable with the observed spectra, i.e., the rupture hits the barrier with high enough energy. For slower ruptures, the spectra are weaker than the observed ones (negative bias), while the faster ruptures result in too strong spectral radiation (positive bias). In addition, there is a small (negative) trade-off between the V_r and the stress drop. However, in the heterogeneous model found by the Bayesian inversion, the high mean rupture speed was not necessary as the rupture includes small-scale accelerations and decelerations over the whole evolution and thus permits smaller mean rupture velocities.

We note that the barrier and heterogeneous models differ in radiation efficiency (~ 0.2 and ~ 0.3 for the heterogeneous and barrier models, respectively). Unlike the rupture velocity, this

difference is not as distinctive, as the value 0.3 lies in the upper bound uncertainty of the marginal posterior PDF of the Bayesian inversion.

As noted by the reviewer, the smooth models presented in the supplement do not exhibit enough stopping phases due to the weak barrier and the consequent gradual rupture arrest. Based on this simple parametric study, this can be improved only by models with ruptures propagating fast enough. On the other hand, generally heterogeneous source models have the capability of incorporating multiple deceleration and acceleration episodes at various scales without the mean rupture velocity being so high.

The barrier model can thus be considered as a complementary approach to our heterogeneous-up-to-small-scale parameterization, as we cannot discern the true rupture velocity based on the amplitude spectral data. Although not a primary point, we believe that our study can contribute to the discussion on whether small-scale heterogeneity is necessary. However, addressing this point properly requires an in-depth study including an exhaustive parametric study of the barrier model and additional data sensitive to rupture velocity, and thus, we keep it for future work.

We keep the gradually arresting smooth model examples in the Supplement as example models corresponding to the ensemble-averaged smooth model. The new comment to Fig. S2 now reads: “In Fig. S2, we show two representatives of smooth models of the directive event, one with symmetric and one with asymmetric rupture propagation. The rupture gradually arrests at the edges of the elliptical source region due to the prescribed weak barrier represented by decreased initial stress and increased strength (both 2 times with respect to the conditions within the rupture). Both models underestimate the AS at frequencies $> \sim 5$ Hz. Our symmetrical model does not capture the azimuthal dependence of the AS. Interestingly, even the asymmetrical model exhibits unexplained azimuthal dependence, confirming the sensitivity of the synthetic AS to the rupture evolution. Note that prescribing a stronger barrier with fast rupture propagation would result in omega-squared spectra due to the radiation from the sudden stop of the rupture in some directions, as shown in the classical models of Sato and Hirasawa (1973) and Madariaga (1976) and extensively studied by Kaneko and Shearer (2015). Although such parameterization warrants further testing against real data, it would require additional data, e.g., data sensitive to rupture velocity, which is beyond the scope of this work. Here, we employ parameterization with generally heterogeneous dynamic parameters, which naturally incorporates multiple deceleration and acceleration episodes at various scales (see Results).”

We have also changed the section title to “Model parameterization and Bayesian inversion”. Moreover, we rewrote several sentences in the text. In Abstract, sentence “We find that the heterogeneity in the rupture propagation down to the smallest scales (~ 100 m) is necessary to fit the high-frequency radiation.” has been updated to “In our models, the heterogeneity down to the smallest scales (100m) is required to fit the observed high-frequency radiation.” In Conclusions, we have changed text “A significant result is that the approximately omega-squared apparent source spectral falloff can be explained only by heterogenous ruptures with fractal properties, in contradiction to smooth models with faster spectral decay.”, for “The approximately omega-

squared apparent source spectral falloff observed empirically can be explained by heterogeneous ruptures with fractal properties.”

The mean spectral bias between observed data and the MAP results (Figure 3a) indicates that low-frequency (<0.8 Hz) energy is systematically missing in the MAP model. Since the vertical axis is logarithmic, the discrepancy reaches approximately 30% at 0.5 Hz, which is substantial given that this is the most resolvable frequency range. Please provide an explanation for this discrepancy and justify the difference.

The underestimation is below the corner frequency and thus related to the seismic moment M_0 . Below, we show the bias for a model from our ensemble with the highest M_0 ($3.5e15$ Nm vs. $2.0e15$ Nm of the MAP model shown in Fig. 3a), demonstrating the consequent overestimation at 0.8 Hz by a similar amount (~30%). The corresponding M_w values are 4.3 and 4.13 for the largest and the best model, respectively. Recalling that the prior constraint was 4.2 ± 0.1 , the models fit the prior constraint similarly well. This apparent discrepancy (similarly to other frequencies) is covered by the estimated uncertainty of the inverted models. We have added a note to the end of the Best rupture model section: “Note that the ~30% underestimation at the lowest frequencies is related to the seismic moment M_0 ; models with higher M_0 overestimate the data by a similar amount in that frequency range.”

The authors state (Line 223) “We fit Eq. (1) using L2 minimization in logarithms to the synthetic MR spectra from our ensemble, considering scalar moment M_0 as known.” Please include the fitting results, as they are crucial for assessing how well Eq. (1) fits across a broad frequency range. Additionally, are there any biases between estimated f_c and the goodness of fit?

Here, we show the bias between the moment rate (MR) spectra from the inversion ensemble (Fig. 4a) and the fitted Brune spectrum:

Thick black curve corresponds to the MAP model. Thick and thin green curves are the ensemble average and one standard deviation. We have added the plots as new Fig. S3.

Note that while the directive Mw4.2 event has an omega-squared source spectrum, the bias is close to zero, suggesting the omega-squared spectrum is a suitable approximation. Conversely, the spectrum of the nondirective Mw4.5 event decays faster, and thus, the omega-squared spectrum overestimates the MR spectrum up to ~ 2 Hz, whereas it underestimates it at higher frequencies. This is indeed reflected in the corner-frequency vs. goodness-of-fit plot shown below, where the misfit is generally higher for the non-directive Mw4.5 event. Despite this, the corner frequencies and the Madariaga stress drop estimates result in correct estimates.

The following figures show the estimated corner frequency vs M_0 , shaded by misfit (L2 norm in \ln):

Except for the possible trade-off between f_c and M_0 , there is no clear trade-off between f_c and misfit. However, the lighter misfit colors clearly demonstrate the generally larger misfit values of the nondirective Mw4.5 event.

We have added a comment on the spectral fit to the text: “As shown in Fig. S3, the fit is slightly worse for the nondirective Mw4.5 event but only above ~3 Hz (i.e., well above the corner frequency) because its spectral decay is faster than the omega-squared.”

The authors state (Line 255) “This seems to contradict the findings of Yoshida (2019), who showed that stress drop estimates can be almost twice as low when the asymmetry of the rupture propagation is neglected.” More context is needed here, as the line of reasoning is unclear. First, what is meant by “the theoretical model of Yoshida (2019)” (Line 259)? Yoshida (2019) used the model of Dong and Papageorgiou (2003) rather than developing a new theoretical model. Furthermore, Kaneko and Shearer (2015) showed that their asymmetrical rupture model is set up to produce a longer source duration than the symmetrical model, reflecting the smaller k values in the asymmetrical model. Since the present study infers similar source durations for the two events, it is inappropriate to interpret the results using the Kaneko and Shearer (2015) models. Please provide more background before comparing the results to previous theoretical models.

The paragraph mentioned by the reviewer is primarily about the determination of the corner frequency and the stress drop inferred from it, which depends on the value of k . The preceding paragraph compares stress drops estimated from the corner frequency inferred from the *moment rate spectra* of the dynamic models. In that case, the theoretical models (Madariaga) provide correct stress drops. However, when the corner frequency is estimated directly from the *apparent source spectra* (AS), it is biased due to the averaging over the stations. Indeed, the apparent source time function (and thus AS) corresponds theoretically to the moment rate function only for a station located perpendicular to the fault at an infinite distance. For normal faults, such a condition is difficult to meet in a real station configuration. To further demonstrate this, we show in the figure below the comparison of the apparent source time functions (left) and spectra (right, in violet) at all our stations (gray curves) with the moment-rate characteristics for a symmetrical barrier model (as shown in response to Q2 above).

As there are no stations that would be exactly perpendicular to the fault, the average observed spectrum (thick green) is very different from the moment rate spectrum (black). Therefore, it is true that our examples should not be compared with the results of Kaneko and Shearer (2015),

because they performed station averaging over the full focal sphere, which might reduce the discrepancy of the stress drop estimates. In our manuscript, we tried to emphasize that the problem arises only if the corner frequency is estimated directly from the azimuthally-limited data (AS) instead of the moment rate spectra. However, the moment rate can be acquired from the data only by the final-fault inversion. Note also that although the source duration of the two events is similar, the events differ in magnitude, and thus the relatively longer duration of the smaller event can be assigned to the rupture asymmetry, consistent with Kaneko and Shearer.

We are sorry for our confusing discussion that mixes the various corner frequency estimates and their applicability based on the symmetry of the rupture. We have simplified some parts of the section and the Fig. 7 caption to distinguish direct and indirect corner frequency (and stress drop) estimates. As suggested by the reviewer, we have also revised the comparison with the published works. Hopefully, the text is now clearer and more precise:

“We also calculate the Madariaga stress drop using the corner frequencies inferred directly by fitting the GIT AS to the omega-square model (shown as arrows in Fig. 7). The stress drop is overestimated significantly for the directive Mw4.2 event, resulting from the overestimated corner frequency. We attribute this to the directivity of the rupture model and the insufficient averaging over the focal sphere (Kaneko and Shearer, 2015). This suggests that the simple spectral stress drop estimates from averaged AS (provided physics-based models, such as Madariaga, are utilized) may be biased in practice, especially for events with strong azimuthal variability of the AS. Therefore, the practitioners should first examine the variability of the AS shapes as their large spread may indicate a directive event for which the standardly inferred source properties might be significantly biased (Fig. 4a). A potential improvement may be in such cases represented by fitting apparent (station-dependent) corner frequencies estimated from the AS using asymmetrical rupture models (e.g., Yoshida, 2019). However, the most reliable approach, regardless of the event’s directivity, is to derive directly the moment rate functions by, e.g., dynamic or kinematic finite-fault inversion (Galović et al., 2025).”

The authors state (in Line 293) that the inferred k^{-2} decay of PSD corresponds to k^{-1} decay of the amplitude spectrum, and such a decay was considered for the initial stress in previous theoretical studies. However, the previous studies (e.g., Ripperger et al, 2007) assumed that stress heterogeneity is described by a power law decay of its wave number spectrum. I might be missing something here, but Figure 8 shows increasing amplitude with wavelength, and hence, the trend is opposite from those assumed in the previous studies.

The inferred spectral decay is indeed k^{-2} , i.e., the spectral amplitudes decrease with increasing wavenumber (=decreasing wavelength). The confusion probably stems from plotting the x-axis (wavelengths) in Fig. 8 in a reverse manner. To avoid the confusion, we have added a note that the wavelengths are shown “plotted as a function of wavelength (the inverse of wavenumber k), displayed from long to short wavelengths (i.e., decreasing from left to right)”. We have also added the term “power law decay” to the Fig. 8 caption and text where appropriate.

Minor comments:

Line 19: The statement “Heterogeneity in the rupture propagation down to the smallest scales is necessary to fit the high-frequency radiation” could be misleading, as it does not consider the debate on whether high-frequency energy arises from elastic heterogeneities near the source rather than rupture propagation. Since this study does not account for elastic heterogeneities, the statement should be revised to avoid confusion.

We have corrected the statement to “rupture heterogeneity” to avoid confusion.

Line 35: Yoshida (2024) should be corrected to Yoshida (2019).

We are sorry for the wrong reference. The correct one is Yoshida (2024) in SRL. We have corrected it in the reference list.

Line 45: The phrase “can attain different than” is unclear. Please clarify.

We have modified the sentence to “the empirical spectra exhibit decay at high frequencies that is different from the omega-squared decay”.

Line 73: The claim that this study performs “an inversion of AS free of path and site effects” seems overstated. Complete removal of path and site effects is generally impossible, particularly due to elastic heterogeneities near the source (e.g., fault damage zones within a few hundred meters). Consider revising this statement.

We agree and have corrected the expression to be more appropriate: “AS corrected for path and site effects.”

Line 90: The phrase “described in Text S1” is inappropriate for the Introduction section, as it references results before they are presented. Move this sentence to the Results section.

We have removed the phrase from the Introduction, keeping the first reference to Text S1 in Results.

Line 145: “due to the prescribed weak nucleation”. This is not explained anywhere.

We agree that the nucleation specified only in Tab. 1 was insufficient. Therefore, we have added a text specifying the nucleation conditions and reference to Table 1 to the Methods section: “We prescribe a weak nucleation with a radius of 0.2 km and a maximum mean overstress of 1 MPa (Tab. 1) to avoid a strong effect on the synthetic AS at high frequencies.”

Line 458: The authors should discuss numerical resolution in terms of cohesive zone sizes (e.g., Day et al., 2005), as is common in dynamic rupture simulation studies.

Since the models are strongly heterogeneous, cohesive zone size varies, and their estimates do not have to be precise. Nevertheless, we have tested the modeling by densifying the FD spatial (and temporal) grid steps to half (5 m, 0.2 ms). For the MAP models, we provide a comparison of the MR spectra and modeling bias below:

The misfit value changes only by 2-4% for the finer sampling, which is negligible considering the accepted models' misfits vary by up to 100%. The dynamic simulations are thus numerically robust. We have added the following text to Methods (section Dynamic rupture simulation and synthetic AS calculation): “We tested the numerical convergence of the dynamic rupture simulation by decreasing the spatial and temporal grid size to half (5 m, 0.2 ms) for the MAP models. The models differ by 2-4% in their AS misfit, which is negligible considering the frequency range up to 25 Hz.”

Reviewer #2:

This manuscript considers dynamic rupture inversion for earthquakes of moment magnitude 4. Dynamic rupture models are essentially a research frontier in understanding the physical processes of earthquake rupture. These models permit more detailed insight into physical processes than kinematic models, spectral models, or point source models. For example, the physical constraints/correlations build into these models often ensure more physically meaningful results. To my knowledge, only few works exist that characterize sources with kinematic models at these magnitudes. After the recent success of dynamic rupture studies, extension to lower magnitudes is quite desirable.

The authors choose a probabilistic framework for their new method which considers apparent-spectra data. In my view, this is attractive since (1) it permits intrinsic uncertainty quantification, (2) permits general parametrizations since the method is nonlinear/numerical in nature, and (3) in some ways future proofs the method since it can be extended because of (1) and (2). There is significant new knowledge to be gained if the methods presented in this manuscript were to be applied more broadly.

I think this is an excellent manuscript and should be of interest to the entire earthquake community and likely beyond. I recommend considering publishing it after addressing the comments below.

Specific comments:

1) The method employs AS for data, and these are “corrected” for certain effects by an inversion process (termed GIT). Since Bayesian methods generally rely on some form of data error estimation or assumption. However, regardless of the assumptions made, the errors need to be preserved in some meaningful way by the pre-processing, here including GIT. Is my understanding correct? Could the authors comment on this or explain in a bit more detail which impacts the GIT has on the overall results?

GIT employs an L2 minimization of the smoothed Fourier Amplitude Spectra into individual terms (site, path, and source). It does so under the assumption of uncorrelated, homoscedastic errors, but without explicitly accounting for measurement uncertainty. In principle, it could use an estimate of the data covariance matrix to obtain the posterior uncertainty of individual terms. However, to the best of our knowledge, the current implementation of GIT, including its Python-based version (GITpy; available at <https://gitlab.rm.ingv.it/inversion/gitpy>), does not incorporate a data covariance matrix in the inversion process. Therefore, it does not provide formal posterior uncertainty estimates for the individual inverted terms, including the source spectral term. In addition, our data also include the GIT residual term, which contains all (non-systematic) effects of source (e.g., directivity), path (e.g., 3D structure), or site (e.g., site effect anisotropy). Although the source effects seem to dominate this term (as seen for the directive event), it is difficult to properly assess the uncertainty related to the other remaining parts. Considering this, we made an assumption on the apparent source spectra uncertainty and tried to explain it in Text S2.

Another possibility would be to assess the AS error based on the variability of the low-frequency part, where the spectra should theoretically converge to a point source estimate. We have added a new explanation of our choice of the data error based on the data standard deviation at low frequencies, where all the AS should collapse, to Text S2. We have added a new Fig. S12 showing the standard deviation (i.e., measure of variability with respect to the average spectrum) as a function of frequency, see below:

2) Figure 1: The figure is generally presented well. A small criticism: I found that the color coding for station azimuth is not clear, in particular w.r.t. azimuthal gaps. Could a rose diagram or similar be included to see the azimuthal distribution more clearly? Alternatively, one could state the max azimuthal gap (I could not find it in the manuscript).

We are sorry for the confusion in Fig. 1 that resulted from using a similar color palette in the a and b panels. The stations in a) are color coded by their AS variations and not by their azimuth, while in b) the spectra were color coded by the corresponding station azimuth. To avoid further confusion, we have modified the color palette for Fig. 1a. In addition, we have added rose histograms to Fig. 1a as insets to show the azimuthal coverage and have modified the caption accordingly. The maximal azimuthal gaps for directive and nondirective events are 16° and 35° , respectively. We include the comment on this in the text:

“The azimuthal coverage is good with 16° and 35° gaps for the directive and nondirective events, respectively, yet with a prevalence of stations in the NW-SE direction (see the rose diagram in Fig. 1a), following the orientation of the Apennines.”

3) The rupture is parametrized on a 125-m grid (and there is a second 10-m for FD calculations that is irrelevant for my question). How is this grid size chosen? Does the result depend on this choice? Does the dynamic model require any form of regularization? I have worked on the effects of discretization choice (with trans-dimensional parametrizations and with regularization) and generally found that model selection is quite important for the results. It would be useful to briefly discuss the parametrization choice and whether regularization or some other form of model selection is useful here.

The 125-m grid was chosen as a compromise based on the corresponding wavelength (for $V_s=3.5$ km/s, it corresponds to $f_{\max}=28$ Hz) and on our previous experience: Valentová et al. (2021) generated a dynamic scenario database and found that the moment rate spectra decayed faster above 2 Hz, which corresponds to the wavelength of their employed model grid (1.2-1.4 km). Therefore, for coarser grids, the rupture heterogeneities might not be sufficient to generate omega-squared radiation in the frequency range (up to 25 Hz) considered. On the other hand, parameters on a finer grid would be unconstrained by the AS data, and thus would require AS extended above 25 Hz to infer the possible k^{-2} power law signature of fractal properties down to such lower scales. In addition, a denser grid would significantly increase the number of model parameters, making the posterior sampling by the MCMC insufficient. The choice of the grid thus serves also as a regularization of the inversion. Another implicit regularization of the kinematic rupture parameters is represented by the dynamic modeling itself (elastodynamic equation and friction law). We have added the following text on this issue: “The parameters are defined on a grid of control points 125 m apart, from which they are bilinearly interpolated onto the finer FD grid. The grid size was chosen so that the reliably resolved maximal simulated frequency corresponds to the maximum frequency of the employed data, 25 Hz (compare with, e.g., Valentová et al., 2021). For more details, see, e.g., Gallovič et al. (2019a,b).”

4) Further on the use of GIT: The goal to remove path and station-side effects. This seems a bit too good to be true, in particular at frequencies up to 25 Hz. Are the residual errors mentioned on line 419 in some way accounted for in data error covariance estimation?

We discuss the data error estimation in relation to the residual error in Text S2. Moreover, we have added a reference to Text S2 into this part of the manuscript. See also our reply to question 1.

5) The methods section (L428 ff) mentions a combination of GIT and BayesISOLA software to constrain inputs for the subsequent inversion. Is there any type of uncertainty propagation in this workflow? It was unclear to me if centroid moment tensor results are used in a probabilistic sense or as point estimates. A similar mention is on L 481; is M_w constrained according to the posterior from BayesIsola or some arbitrary symmetric distribution (there is a mention of plus/minus 0.05)?

Indeed, the centroid as well as hypocenter location uncertainties estimated from BayesIsola are not part of the Bayesian dynamic source inversion, i.e., they are not propagated to the uncertainty of the source properties. The only exception is M_0 estimated from BayesIsola, which serves as a prior mean value. We assume a Gaussian PDF in M_w , which corresponds to a log-normal PDF in M_0 (see Fig. 6), with a conservative error estimate of 0.05, as the reviewer

mentioned. As shown in Fig. 6, the dynamic source inversion provided a smaller posterior uncertainty in this parameter.

Considering the fault plane orientation (strike-dip-rake), we also performed a simple synthetic test, where we assumed an incorrect fault orientation by 7 degrees in Kagan angle, one of the BayesIsola results. The effect on the apparent spectra is shown as a spectral bias below and demonstrates that the effect is rather random and more significant at frequencies above the corner frequency. Furthermore, we assume that there are various other effects at these frequencies, such as fault plane nonplanarity, contributing to the variability of apparent spectra that are difficult to determine. Therefore, we neglected the uncertainty in fault plane orientation in the dynamic source inversion for now.

We thank the reviewer for the suggestion. We admit that neglecting the centroid variability and location uncertainty is a simplification that can potentially be improved in further applications. We mention it in the newly added Text S3 about the method limitations (see further).

Minor comments:

1) L45: Wording unclear w.r.t. “attain”

We have modified the sentence to “Empirical source spectra often exhibit high-frequency decay that deviates from the classical omega-squared model.”

2) L71: functions (plural)

Corrected.

3) S2: 5,600 and not 5.600

We have corrected it.

4) Overall, some editing for English is required. I think I understood the meaning of the text most of the time but both manuscript and supplement will require editing for many small issues as in the previous comments.

We used Grammarly to check the English grammar in the original version. In the revised manuscript, we have employed AI to improve the English of several sentences in the main text and Supplement.

5) The setup of PT (L515 ff) appears odd to me. Randomly assigning temperatures between 1 and 100 seems unlikely to result in good acceptance rates for swaps. Were acceptance rates for temperature swaps considered in tuning the algorithm? In my experience this can make an enormous difference and I have never seen an approach similar to the one described here.

We consider our choice reasonable because $T=100$ corresponds to a 10x higher data error. Since the data error selection is ambiguous (see Text S2 in the Supplement), especially due to data inter-frequency correlation, we think the assumed data error range is rather conservative.

The acceptance rates in the chains of our dynamic inversion are mainly affected by the parameters of the dynamic models themselves. For example, a perturbation close to the nucleation region generates models that are discarded (not accepted) due to differences in magnitudes. We admit that we tuned the algorithm mainly considering these perturbations and paid less attention to the temperature swapping, which should be improved in the future for better effectiveness of the posterior PDF sampling.

6) L521: The chain thinning applied here should be noted to not positively affect the posterior ensemble. Rather, it is merely applied to manage storage size. The full posterior (all samples, regardless of correlation) always produce better results.

We thank the reviewer for this suggestion. We admit that it was introduced to reduce the pseudoreplication of the samples because the correlation between the accepted dynamic models is usually high. We can avoid chain thinning in our future applications. As suggested, we have corrected the sentence to read: "...keeping only every 10th accepted model to manage the storage size of the output models."

I hope you find these comments useful and constructive.

Best wishes, Jan Dettmer (Yukon Geologic Survey / UCalgary)

Reviewer #3:

I read the paper “Physics-based broadband characterisation of weak earthquakes” by Gallic et al. with considerable interest. It is a fine piece of work and appears to be of high quality and completeness. It shows how spatial variations in earthquake amplitude spectra can be used to determine important details about the rupture properties of even relatively small earthquakes, in this case of magnitudes 4.2 and 4.5. It seems remarkable to see rupture directivity resolved for an earthquake as small as magnitude 4.2. Moreover it shows how failing to account for these details can sometimes lead to very significant bias in the conventional methods for estimating important parameters like stress drop. I think the paper is an important step forward in estimating earthquake source parameters.

However, while I myself found the paper very interesting I would have to say that I think it appeals mainly to a narrow audience of ground motion seismologists. If Nature is fine with accepting a paper that may not have broad appeal, then the paper can be published with only minor revision. If instead it is Nature’s goal to publish only papers with broad appeal, then this paper may not be such a good fit. In order to make it appeal to a wider audience I think it would require major revision, such as perhaps considering a much larger dataset - difficult, given that these two events each took 30 days of run time.

1) My main substantive comment is that in the section on “GIT derived apparent spectra”, some important assumptions are mentioned, such as the value of the high-frequency attenuation coefficient κ ($=0.015$, Line 411), and the use of random perturbations in the strike, dip, and rake to mimic random perturbations to crustal velocity structure. These all may be valid assumptions, but without further evidence. I’m left wondering whether there could be trade-offs between some of the source properties and the κ value. Random crustal velocity perturbations will not only obscure the S-wave radiation pattern but can also produce frequency dependent apparent attenuation due to scattering. Could such effects be mapped into the source properties if they are not accounted for? In any case, I feel more discussion of these is warranted.

We understand the concerns of the reviewer. Accounting for all these effects introduces a systematic change in all spectra over the whole frequency range. These changes can affect the estimation of magnitude at low frequencies or small-scale source characteristics at high frequencies. For example, the effect of various values of κ , corresponding to high frequency attenuation of the carefully selected reference sites, is shown in the figure below:

The smaller kappa value reduces the spectral decay at high frequencies and can thus result in more pronounced small-scale rupture characteristics. Conversely, the higher kappa value might have the opposite effect.

Discussing with other practitioners who use these types of spectral data to derive source properties, we understand that there is no certain evidence on which parameters are the most appropriate. Therefore, we tried to account for the site amplification by combining all the effects to be consistent with other studies using spectral data. For example, our value of kappa was selected to be consistent with rock site conditions in Central Italy (Lanzano et al., 2022; Morasca et al., 2023). Further research might help to improve the characterization of the residual spectral amplification and thus improve the source parameter estimates. We expand the discussion on the possible trade-offs that can be introduced by these effects in new Text S3.

2) As with all Bayesian inversions, an important question is: Has it converged? With the sampling of 250k models over 30 days run time, I could well believe the solution has converged. However, some indication of how the authors assessed this, such as examining the character of the chain histories of likelihood values, should be discussed.

We calculated the Gelman-Rubin criterion (Gelman and Rubin, 1992) for several mean parameters (slip, stress drop, or energy estimates). We did not use any particular model parameter to track the convergence, because the dynamic parameters may significantly differ between the individual models due to the missing phase information in the data. Histograms for the mean stress drop over individual MPI jobs (pseudochains, in colors) for cold chains ($T=1$) are shown below, demonstrating the differences between the pseudo-chains:

Based on the Gelman-Rubin criterion for the mean parameters, most of the chains converged with values <1.1 . Only the values of the mean stress drop were a little higher, 1.14 for the directive event. For the non-directive event, we found higher Gelman-Rubin values for stress drop 1.12 and rupture area 1.19.

Line 519: Shouldn't the total number of models visited by 5k steps x 8 chains x 6 jobs = 240k, not 250k?

Our calculation is only approximate. Since the number of steps was larger than 5k, we have modified the sentence as follows: "...each chain performed more than 5,000 steps, i.e., about 250k models were visited."

Figure 1: It can be difficult to make out the stars that denote the epicentres. Could they be made bigger, or a different colour so they'd stand out more? The lines connecting the beachballs to the respective epicenters are also very thin.

As suggested, we have increased the size of the star (epicenter) and made the line thicker. However, we have kept the epicenter's black color because we believe it is now sufficiently visible.

Figures 6 & 7: These are a little confusing because the units of the x-axis at the bottom of the figure are actually given as part of the figure title, at the top of the figure. I recommend moving the units to the bottom of the figure, just below the x-axis, as has been done with other figures.

We understand the potential confusion, but including the units in the titles was chosen to save space in the plots, i.e., to keep the individual panels sufficiently large. Note that we also use the same style in other figures (2, 5, 7) and in our previous publications. If acceptable, we would rather retain the present style.

We thank the reviewers for the provided comments. We have addressed them as we describe below. Note that we have also made some minor improvements to the text.

Reviewer #1 (Remarks to the Author):

I appreciate the authors' efforts to revise the manuscript and provide detailed responses to all comments. The study continues to offer a promising approach to dynamic rupture inversion for small earthquakes, and I acknowledge the added value of the new parametric study shown in the rebuttal. However, I am not satisfied with the revision in its current form.

My primary concern remains the same as in my initial review. I feel that the authors have not adequately addressed the first major comment of Reviewer #1 regarding the role of abrupt rupture arrest models. While I acknowledge that additional simulations were conducted and shared in the rebuttal letter, the results of this important analysis are not included in the revised manuscript or its Supplementary Information. Presenting such key results only in the rebuttal is inappropriate, as it prevents readers of the published paper from critically evaluating the comparison between abrupt arrest and heterogeneous rupture models.

The revised manuscript continues to emphasize the necessity of small-scale heterogeneity to fit the observed high-frequency spectra. This is a central conclusion of the paper and, if correct, would represent a significant advance. However, without properly documenting the performance of sudden arrest models—beyond the gradual-arrest smooth models shown in the SI—the claim that heterogeneity is required remains insufficiently supported. The authors themselves acknowledge that barrier models with fast rupture can produce omega-squared falloff and fit the observed spectra reasonably well under certain conditions. Yet this important nuance is hidden in the rebuttal and omitted from the actual manuscript.

If the barrier models performed significantly worse than the heterogeneous models, then the comparison should be documented and discussed explicitly in the paper. If the differences are more subtle—perhaps depending on trade-offs with rupture speed or stress drop—then that too should be part of the discussion. In either case, the reader must be given access to these results in order to independently evaluate the authors' interpretation.

We agree that some of our statements regarding the rupture heterogeneity were confusing. Indeed, in our study, the heterogeneous distribution of parameters is an assumption, and what we infer is the power-law properties of the parameters. We are unable to disprove that the other types of models are incorrect, but we consider our parameterization more realistic when fitting real data than the smooth ruptures, which are undoubtedly important from the theoretical point of view. We also agree that performing an appropriate inversion with the smooth barrier models would be interesting, but it cannot be fully incorporated in our manuscript as it requires a devoted in-depth study.

I also note that the revised version makes several definitive statements implying that heterogeneity is required (e.g., Abstract, Lines 21–22: "In our models, the heterogeneity down to the smallest scales (~100 m) is required to fit the observed high-frequency radiation."). In light of the alternative modeling presented in the rebuttal, such statements are too strong without presenting the full scope of tested models.

We have modified the Abstract to clearly distinguish between our parameterization (heterogeneous) and the properties required to fit the data (power-law spectra).

Therefore, I encourage the authors to revise the manuscript by:

(i) Incorporating the abrupt arrest (barrier) model results into the manuscript or SI in a clearly labeled section, including key figures and discussion of misfits, parameter trade-offs, and spectral comparisons.

We were very cautious to present the parametric study of the barrier models within the paper (and keeping it public only as a rebuttal), as it provides a very limited number of manually set simulations compared to the McMC inversion that visited 250k models. Therefore, it cannot be confronted with the results of the Bayesian inversion in the main text. In addition, the spectral properties of such models have already been properly studied, e.g., Kaneko and Shearer (2015), which we have referenced. Based on the suggestion, we have added the tests presented in the previous rebuttal letter to the SI as a new Text S4. We refer to this text when clarifying our parameterization in the Materials and Methods.

(ii) Clarifying the framing of their conclusions regarding the necessity of heterogeneity in light of the alternative models.

We have modified the Abstract and other statements to clearly express that our inversion infers the power law for the assumed heterogeneity.

(iii) Moderating claims in the Abstract, Results, and Conclusions to reflect the fact that other models (with abrupt arrest) can also explain aspects of the high-frequency radiation, albeit with limitations or different trade-offs.

In all relevant parts, we modify the text to clearly distinguish our assumption (heterogeneity) from the inferred properties (power-law). We also mention the smooth barrier models as an alternative, but we prefer heterogeneity as a more realistic description of the underlying complex faulting process, which we support by adding references.

These additions would substantially improve the rigor, transparency, and impact of the paper. Until these revisions are made, I would not consider the current manuscript suitable for publication in Nature Communications.

Minor comment:

Line 62: The authors mention that “Yoshida (2019) demonstrated that asymmetrical rupture propagation may result in biased stress drop estimates if the directivity effect on the source spectra is neglected.” It could be helpful to also acknowledge that methods such as the second moment approach (e.g., McGuire et al., 2017; McGuire and Kaneko, 2018) offer an alternative pathway to improved stress drop estimates by better resolving source geometry and rupture propagation direction.

We agree that another way of improving the stress drop estimates is by constraining the fault size instead of the corner frequency, following also the simulation analysis in Gallovič and Valentová (2020). However, such an estimate by, e.g., the second moment approach will also be subject to uncertainty, which will be translated into uncertainty of the stress drop. We have added a note together with new references to the Discussion part, where we discuss the potential bias in stress drop estimation in detail.

Reviewer #2 (Remarks to the Author):

I would like to thank the authors for an exceptionally thorough response to my comments. I have no further concerns with this manuscript and recommend it be considered for publication. It's excellent work in my opinion.

Review of Manuscript
“Physics-based broadband characterization of weak earthquakes”
by František Gallovič, Sara Sgobba and L'ubica Valentová K.

This manuscript considers dynamic rupture inversion for earthquakes of moment magnitude 4. Dynamic rupture models are essentially a research frontier in understanding the physical processes of earthquake rupture. These models permit more detailed insight into physical processes than kinematic models, spectral models, or point source models. For example, the physical constraints/correlations build into these models often ensure more physically meaningful results. To my knowledge, only few works exist that characterize sources with kinematic models at these magnitudes. After the recent success of dynamic rupture studies, extension to lower magnitudes is quite desirable.

The authors choose a probabilistic framework for their new method which considers apparent-spectra data. In my view, this is attractive since (1) it permits intrinsic uncertainty quantification, (2) permits general parametrizations since the method is nonlinear/numerical in nature, and (3) in some ways future proofs the method since it can be extended because of (1) and (2). There is significant new knowledge to be gained if the methods presented in this manuscript were to be applied more broadly.

I think this is an excellent manuscript and should be of interest to the entire earthquake community and likely beyond. I recommend considering publishing it after addressing the comments below.

Specific comments:

- 1) The method employs AS for data, and these are “corrected” for certain effects by an inversion process (termed GIT). Since Bayesian methods generally rely on some form of data error estimation or assumption. However, regardless of the assumptions made, the errors need to be preserved in some meaningful way by the pre-processing, here including GIT. Is my understanding correct? Could the authors comment on this or explain in a bit more detail which impacts the GIT has on the overall results?
- 2) Figure 1: The figure is generally presented well. A small criticism: I found that the color coding for station azimuth is not clear, in particular w.r.t. azimuthal gaps. Could a rose diagram or similar be included to see the azimuthal distribution more clearly? Alternatively, one could state the max azimuthal gap (I could not find it in the manuscript).
- 3) The rupture is parametrized on a 125-m grid (and there is a second 10-m for FD calculations that is irrelevant for my question). How is this grid size chosen? Does the result depend on this choice? Does the dynamic model require any form of regularization? I have worked on the effects of discretization choice (with trans-dimensional parametrizations and with regularization) and generally found that model selection is quite important for the results. It would be useful to briefly

discuss the parametrization choice and whether regularization or some other form of model selection is useful here.

- 4) Further on the use of GIT: The goal to remove path and station-side effects. This seems a bit too good to be true, in particular at frequencies up to 25 Hz. Are the residual errors mentioned on line 419 in some way accounted for in data error covariance estimation?
- 5) The methods section (L428 ff) mentions a combination of GIT and BayesSOLA software to constrain inputs for the subsequent inversion. Is there any type of uncertainty propagation in this workflow? It was unclear to me if centroid moment tensor results are used in a probabilistic sense or as point estimates. A similar mention is on L 481; is Mw constrained according to the posterior from BayesSOLA or some arbitrary symmetric distribution (there is a mention of plus/minus 0.05)?

Minor comments:

- 1) L45: Wording unclear w.r.t. "attain"
- 2) L71: functions (plural)
- 3) S2: 5,600 and not 5.600
- 4) Overall, some editing for English is required. I think I understood the meaning of the text most of the time but both manuscript and supplement will require editing for many small issues as in the previous comments.
- 5) The setup of PT (L515 ff) appears odd to me. Randomly assigning temperatures between 1 and 100 seems unlikely to result in good acceptance rates for swaps. Were acceptance rates for temperature swaps considered in tuning the algorithm? In my experience this can make an enormous difference and I have never seen an approach similar to the one described here.
- 6) L521: The chain thinning applied here should be noted to not positively affect the posterior ensemble. Rather, it is merely applied to manage storage size. The full posterior (all samples, regardless of correlation) always produce better results.

I hope you find these comments useful and constructive.

Best wishes, Jan Dettmer (Yukon Geologic Survey / UCalgary)